# TPX2 lactylation is required for the cell cycle regulation and hepatocellular carcinoma progression

Shengzhi Liu[1],*, Jin Cai[2],* , Xiaoyu Qian[1,2], Junjiao Zhang[3], Yi Zhang[1], Xiang Meng[1], Mingjie Wang[1], Ping Gao[1,2,3] , Xiuying Zhong[1,2]

Targeting protein for Xklp2 (TPX2) is critical for mitosis and spindle assembly because of its control of Aurora kinase A (AURKA). However, the regulation of TPX2 activity and its subsequent effects on mitosis and cancer progression remain unclear. Here, we show that TPX2 is lactylated at K249 in hepatocellular carcinoma (HCC) tumour tissues and that this process is regulated by the lactylase CBP and the delactylase HDAC1. Lactate reduction via either shRNAs targeting lactate dehydrogenase A or the lactate dehydrogenase A inhibitor GSK2837808A decreases the level of TPX2 lactylation. Importantly, TPX2 lactylation is required for the cell cycle regulation and tumour growth. Mechanistically, TPX2 lactylation disrupts protein phosphatase 1 (PP1) binding to AURKA, enhances AURKA T288 phosphorylation, and facilitates the cell cycle progression. Overall, our study reveals a previously unappreciated role of TPX2 lactylation in regulating cell cycle progression and HCC tumorigenesis, exposing an important correlation between metabolic reprogramming and cell cycle regulation in HCC.

## Introduction

Lactate generated during enhanced glycolysis in cancer cells serves not only as an energy substrate but also as a signalling molecule and an immunosuppressive factor (Ippolito et al, 2019; Li et al, 2022). Recently, lactate-driven histone lysine lactylation (Kla), a new epigenetic modification, has been reported to affect critical cellular processes related to physiology and disease, such as inflammation (Zhang et al, 2019), vascular function (Wang et al, 2022), fibrosis (Cui et al, 2021), and neuromodulation (Pan et al, 2022). However, how lactylation regulates tumorigenesis by modifying nonhistone proteins remains largely unclear. The acquisition of growth signalling autonomy, which is closely associated with the cell cycle clock, is another hallmark of cancer (Hanahan & Weinberg, 2011).

Dysregulation of cell cycle regulators enables cancer cells to enter the active cell cycle. Post-translational modifications (PTMs), including enzyme-catalysed phosphorylation (Suski et al, 2021), ubiquitination (Rape et al, 2006; Dang et al, 2021), and acetylation (Fournier et al, 2016), are important in modulating the activity of essential cell cycle proteins. However, how lactylation affects cell cycle regulators is unclear.

Targeting protein for Xklp2 (TPX2) is a microtubule nucleation factor that is widely known for its conserved roles in spindle assembly and mitosis (Heidebrecht et al, 1997; Wittmann et al, 2000; Neumayer et al, 2014). TPX2 has also been identified as a mitotic regulator of Aurora kinase A (AURKA) (Kufer et al, 2002; Bayliss et al, 2003). The aberrant expression of TPX2 has been found in various cancers and is closely correlated with overall survival in cancer patients (Wittmann et al, 1998; Wei et al, 2013; Huang et al, 2014; Yan et al, 2018). However, the TPX2 protein structure is largely intrinsically disordered, which poses a challenge for the development of anticancer drugs that directly inhibit TPX2 activity. Transient TPX2 acetylation during early mitosis reduces ubiquitination and stabilizes the TPX2 protein, and dysregulation of TPX2 acetylation and accumulation contributes to mitotic defects (Lin et al, 2018). TPX2 can also be phosphorylated at numerous sites (Neumayer et al, 2014). Therefore, exploring the roles of PTMs in the function of TPX2 may provide new strategies for developing anticancer drugs targeting TPX2.

In the present study, we sought to explore the biological importance of lysine lactylation in hepatocellular carcinoma (HCC) progression by regulating the function of cell cycle regulators and discovered that TPX2 is lactylated at K249 via CBP and HDAC1 in HCC cells. TPX2 lactylation is required for the cell cycle regulation and cancer progression. Mechanistically, TPX2 lactylation prevents PP1 from binding to AURKA and thus enhances its phosphorylation. These findings demonstrated that TPX2 lactylation is a critical mechanism for cell cycle regulation contributing to HCC progression, suggesting that inhibition of the

---

[1]School of Medicine, South China University of Technology, Guangzhou, China    [2]Medical Research Institute, Guangdong Provincial People's Hospital, Guangdong Academy of Medical Sciences, Southern Medical University, Guangzhou, China    [3]School of Biomedical Sciences and Engineering, South China University of Technology, Guangzhou International Campus, Guangzhou, China

Correspondence: pgao2@ustc.edu.cn; zxywawj@ustc.edu.cn
*Shengzhi Liu and Jin Cai contributed equally to this work

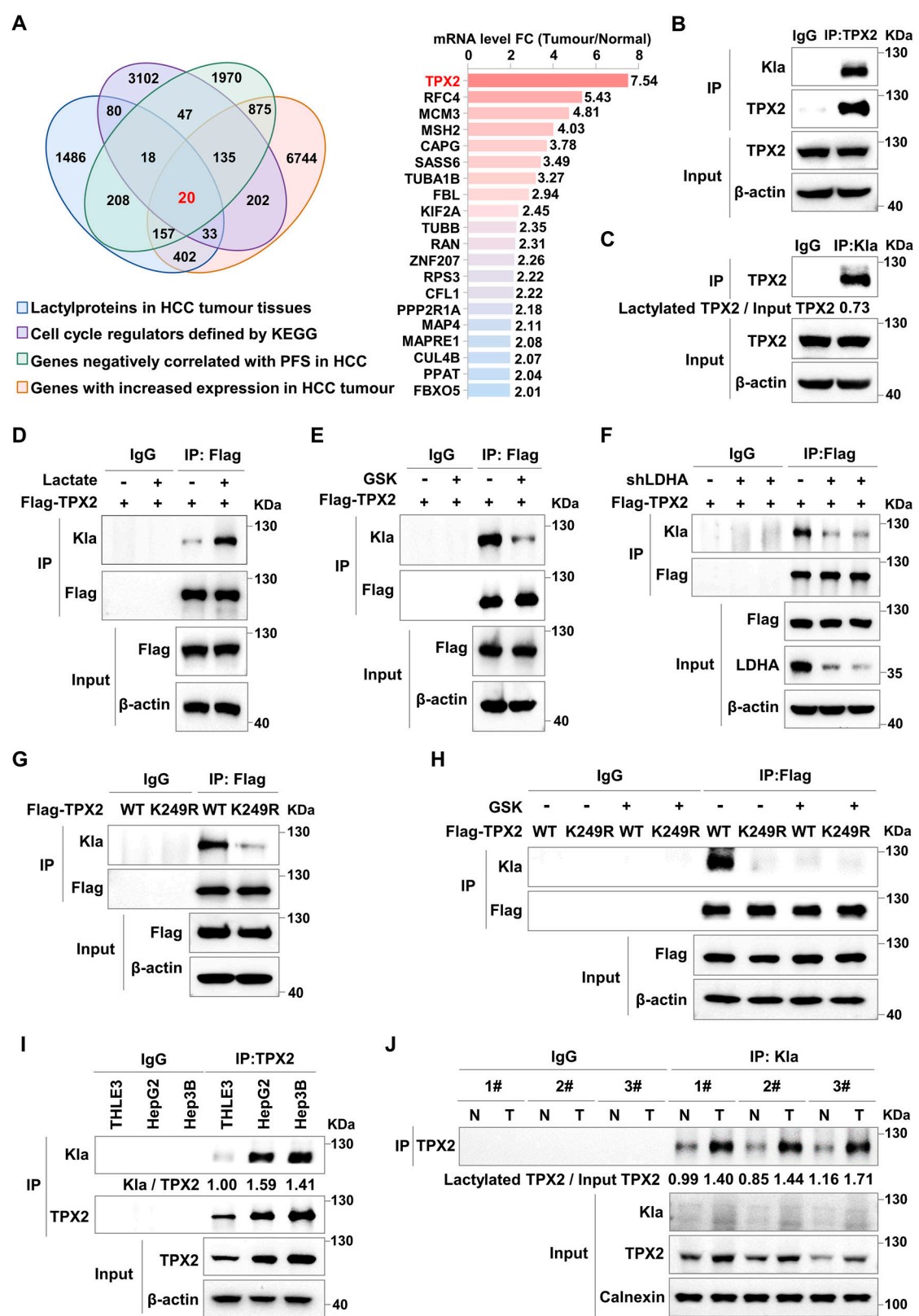

**Figure 1. TPX2 is lactylated at K249 in hepatocellular carcinoma (HCC) tumour tissues.**

**(A)** Venn diagram illustrating the intersection between the four cohorts. Cohort 1 shows the lactylproteins in HCC tumour tissues from lactylation-modified proteomics data from published papers (Yang et al, 2023). Cohort 2 shows the cell cycle regulators defined by the KEGG database. Cohort 3 shows the genes that were negatively correlated with PFS in HCC patients from TCGA database ($P < 0.01$). Cohort 4 shows the genes with increased mRNA expression in HCC tumour tissues compared with those

lactate/TPX2 lactylation/AURKA axis might be a promising therapeutic cancer strategy.

# Results

### TPX2 is lactylated at K249 in HCC tumour tissues

To explore the impact of lactylation on cell cycle progression in cancer cells, we first screened the lactylome of the HCC cohort (Yang et al, 2023), transcriptomic data from HCC patients in The Cancer Genome Atlas, for the key cell cycle regulators that are modified by lactylation and might contribute to HCC progression. The results revealed that TPX2, which is highly expressed in clinical HCC tissues, might be lactylated (Figs 1A and S1A). Moreover, elevated TPX2 expression was significantly negatively correlated with progression-free survival in HCC patients (Fig S1B). Immunoprecipitation (IP) assays with anti-TPX2 or anti-Pan-Kla antibodies in HepG2 cells followed by Western blot analysis with antibodies against Pan-Kla or TPX2 confirmed that almost 73% TPX2 was modified by lactylation in HCC cells (Fig 1B and C). Subsequent IP assays in HCC cells expressing Flag-tagged TPX2 confirmed TPX2 lactylation (Fig S1C and D). We next explored the effect of lactate on TPX2 lactylation. As expected, lactate treatment increased TPX2 lactylation in HCC cells (Fig 1D), and the lactate dehydrogenase A (LDHA) inhibitors GSK2837808A and sodium oxamate had opposite effects (Figs 1E and S1E). Similarly, TPX2 lactylation was decreased in HepG2 cells with LDHA knockdown (Fig 1F). Similar results were observed in HEK293T cells (Fig S1F–H). Taken together, these data revealed that intracellular lactate induced TPX2 lactylation in HCC cells.

Lysine 249 (K249) of TPX2, which is highly conserved in different species, has been shown to be lactylated in the lactylome (Fig S1I). To verify that TPX2 is lactylated at K249, we constructed a vector expressing mutant TPX2 by site-directed mutation of lysine 249 of TPX2 to arginine (TPX2$^{K249R}$). IP assays in HepG2 and Hep3B cells with Flag-tagged WT TPX2 (TPX2$^{WT}$) or TPX2$^{K249R}$ overexpression followed by Western blot analysis revealed that TPX2$^{K249R}$ resulted in dramatically less lactylation than did the TPX2$^{WT}$ (Figs 1G and S1J). In addition, compared with that in the TPX2$^{WT}$ group treated with DMSO, the lactylation of TPX2 in HepG2 cells treated with GSK2837808A or oxamate was reduced to a comparable extent as that in the TPX2$^{K249R}$ group treated with DMSO (Figs 1H and S1K). These results indicated that TPX2 was modified at K249 by lactylation in HCC cells. The function of TPX2 is also regulated by

acetylation, so we sought to determine whether acetylation also occurs at K249 of TPX2. The results showed that the acetylation levels of TPX2 were no different between TPX2$^{WT}$ and TPX2$^{K249R}$ groups (Fig S1L). These data indicated that TPX2 was lactylated at the K249 site.

The higher lactylation level of TPX2 was detected in HepG2 and Hep3B cells compared with that in normal liver cell THLE3 (Fig 1I). To further investigate whether TPX2 is lactylated in vivo, we employed a spontaneous mouse model of HCC by injecting YAP5SA plasmids into mice and detected TPX2 lactylation in HCC tumour tissues in vivo via the IP assay. We observed greater TPX2 lactylation in YAP5SA-induced HCC tissues than in adjacent noncancerous tissues (Fig 1J). These results demonstrated that TPX2 lactylation was induced in HCC cells and tumour tissues, suggesting that TPX2 lactylation might contribute to HCC progression.

### TPX2 is lactylated by CBP and delactylated by HDAC1

To identify the lactylases of TPX2, we screened multiple lactylases, including p300 (Zhang et al, 2019), CBP (Chen et al, 2024a), KAT5 (Jia et al, 2023), KAT8 (Xie et al, 2024), and GCN5 (Wang et al, 2022), in HEK293T cells and found that the lactylation of TPX2 was markedly increased with CBP overexpression (Fig 2A). Moreover, the lactylation of TPX2 in HepG2 cells with CBP knockdown was markedly decreased (Fig 2B). In vitro lactylation assays using Flag-tagged TPX2 and HA-tagged CBP proteins that are purified from HEK293T cells showed that TPX2 was lactylated only in the presence of both CBP and lactyl-CoA (Fig S2A). These results suggested that CBP mediated TPX2 lactylation in a lactyl-CoA–dependent manner. Furthermore, the IP assay confirmed the interaction between TPX2 and CBP in HepG2 cells (Figs 2C and S2B). Similar results were observed in HEK293T cells overexpressing Flag-CBP and HA-TPX2 (Fig S2C). Confocal fluorescence microscopy with GFP-TPX2 and antibody against CBP showed that TPX2 colocalized with CBP in Hep3B cells (Fig S2D). We further detected TPX2 lactylation in HepG2 cells with CBP knockdown and TPX2$^{WT}$ or TPX2$^{K249R}$ overexpression. The results revealed that CBP knockdown resulted in a similar lactylation level of TPX2$^{WT}$ to that of TPX2$^{K249R}$ (Fig 2D). Taken together, our results suggested that TPX2 K249 was lactylated by CBP.

HDAC1-3 and SIRT1-3 were identified as robust Kla erasers (Moreno-Yruela et al, 2022; Jin et al, 2023; Sun et al, 2023; Zhang et al, 2023; Chen et al, 2024b). To identify the delactylases of TPX2, we screened multiple delactylases in HEK293T cells and found that TPX2 lactylation decreased markedly with HDAC1 overexpression

in tumour-adjacent noncancerous liver tissues from TCGA database (fold change > 2). The histogram shows the fold changes in the mRNA levels (tumour/normal) of the 20 intersection genes. **(B)** Western blot analysis of the lactylation of TPX2 in HepG2 cells after immunoprecipitation (IP) assays with anti-TPX2. **(C)** Western blot analysis of the lactylation of TPX2 in HepG2 cells after IP assays with anti-Pan-Kla. Blot bands were quantified by ImageJ, and the fraction of lactylated TPX2 was represented by the ratio of the immunoprecipitated TPX2 band to the corresponding Input TPX2 band. **(D, E)** Western blot analysis of the lactylation of TPX2 in HepG2 cells with Flag-TPX2 overexpression using IP samples as indicated. HepG2 cells were treated with lactate (25 mM) or GSK2837808A (75 $\mu$M) for 24 h. **(F)** Western blot analysis of the lactylation of TPX2 in HepG2 cells with Flag-TPX2 overexpression and lactate dehydrogenase A knockdown using IP samples as indicated. **(G)** Western blot analysis of the lactylation of TPX2 in HepG2 cells with Flag-TPX2WT or Flag-TPX2$^{K249R}$ overexpression using IP samples as indicated. **(H)** Western blot analysis of the lactylation of TPX2 in HepG2 cells with Flag-TPX2$^{WT}$ or Flag-TPX2$^{K249R}$ overexpression using IP samples as indicated. HepG2 cells were treated with GSK2837808A (75 $\mu$M) for 24 h. **(I)** Western blot analysis of the lactylation of TPX2 in the indicated cell lines. Blot bands were quantified by ImageJ, and the fraction of lactylated TPX2 was represented by the ratio of the Kla band to the immunoprecipitated TPX2 band. **(J)** Western blot analysis of the lactylation of TPX2 in tumour-adjacent noncancerous liver tissues (Normal) and liver cancer tumour tissues (Tumour) from the YAP5SA-induced HCC mouse model using IP samples as indicated. Blot bands were quantified by ImageJ, and the fraction of lactylated TPX2 was represented by the ratio of the immunoprecipitated TPX2 blot band to the corresponding Input TPX2 blot band.
Source data are available for this figure.

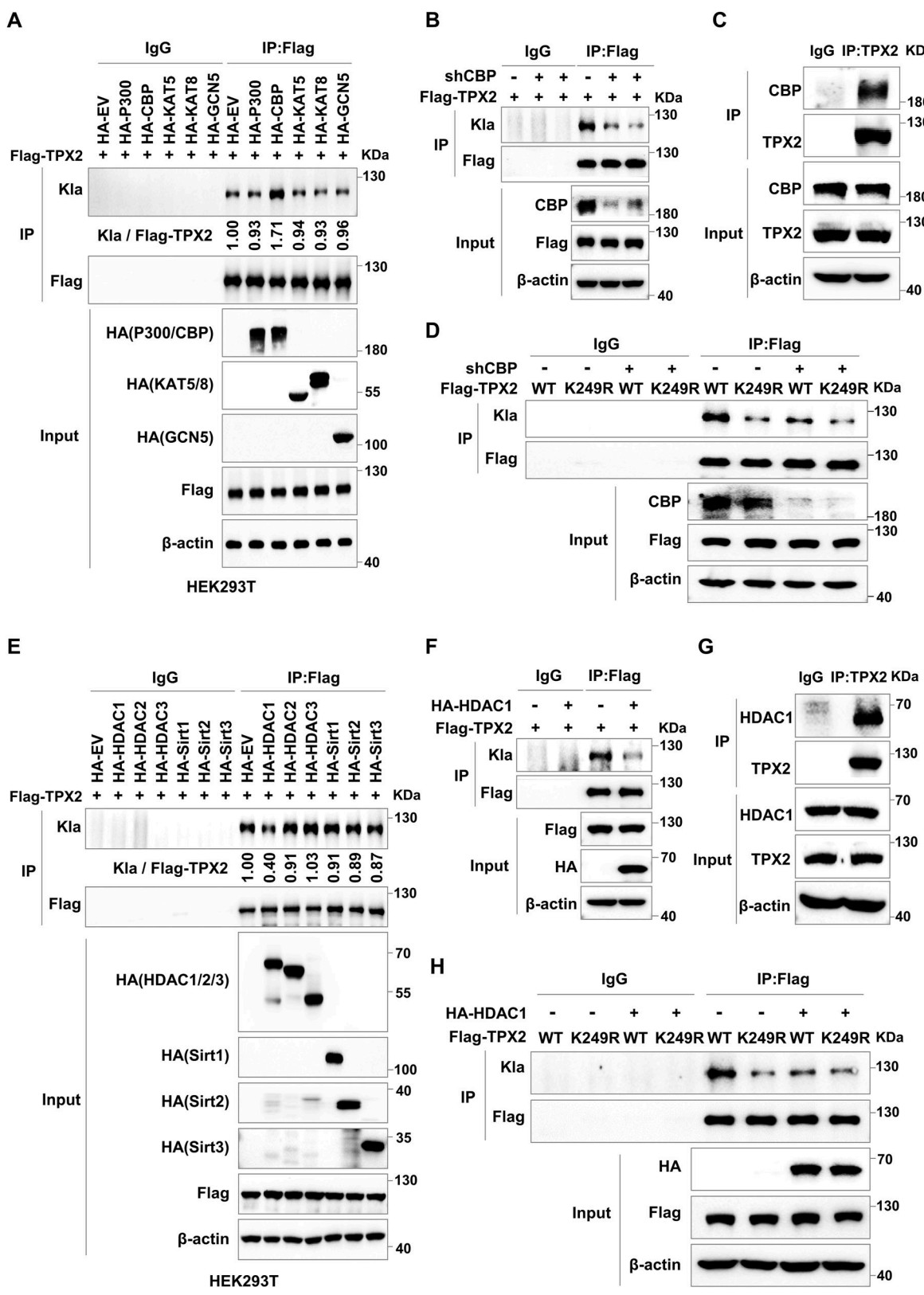

**Figure 2. TPX2 is lactylated by CBP and delactylated by HDAC1.**

**(A)** Western blot analysis of the lactylation of TPX2 in HEK293T cells with Flag-TPX2 and lactylase overexpression using IP samples as indicated. HA blots corresponding to different regions of the filter are shown at the same exposure. Blot bands were quantified by ImageJ, and the fraction of lactylated TPX2 was represented by the ratio of the Kla band to the immunoprecipitated Flag-TPX2 band. **(B)** Western blot analysis of the lactylation of TPX2 in HepG2 cells with Flag-TPX2 overexpression and CBP

(Fig 2E). Similar results were observed in HepG2 cells over-expressing HDAC1 (Fig 2F). These results suggested that TPX2 was delactylated by HDAC1. Moreover, IP assays demonstrated the interaction between TPX2 and HDAC1 in both HepG2 (Figs 2G and S2E) and HEK293T cells (Fig S2F). We also verified the colocalization of TPX2 and HDAC1 in Hep3B cells by immunofluorescence analysis (Fig S2G). In addition, HDAC1 overexpression resulted in a similar lactylation level of TPX2$^{WT}$ to that of TPX2$^{K249R}$ (Fig 2H), which further indicated that HDAC1 is the delactylase of TPX2. In conclusion, our findings revealed that TPX2 was lactylated by CBP and delactylated by HDAC1.

### TPX2 lactylation promotes tumour growth in vitro and in vivo

To further explore the effects of TPX2 lactylation on HCC cell growth, we generated HCC cell lines with endogenous TPX2 knockdown (shTPX2 targeting the 3′UTR of TPX2 transcripts) and Flag-TPX2$^{WT}$ or Flag-TPX2$^{K249R}$ re-expression. We found that TPX2$^{WT}$, but not TPX2$^{K249R}$, restored the retarded cell growth induced by TPX2 knockdown in HepG2 cells (Figs 3A and B and S3A and B). Similar results were observed in Hep3B cells (Fig S3C and D). In addition, this restoration of cell growth via re-expressing TPX2$^{WT}$ in HepG2 cells with endogenous TPX2 knockdown was attenuated by the LDHA inhibitor GSK2837808A (Figs 3C and S3E). In summary, these results indicated that TPX2 lactylation was crucial for the growth of HCC cells.

We next evaluated the effect of TPX2 lactylation on the in vivo HCC tumour growth by employing an HCC xenograft mouse model established via the subcutaneous injection of HepG2 cells with endogenous TPX2 knockdown and Flag-TPX2$^{WT}$ or Flag-TPX2$^{K249R}$ re-expression. Re-expression of TPX2$^{WT}$, but not TPX2$^{K249R}$, could largely restore HepG2 xenograft growth suppressed by endogenous TPX2 knockdown (Fig 3D and E), although the protein level of TPX2 in the TPX2$^{K249R}$ group was equal to that in the TPX2$^{WT}$ group (Fig 3F). These results indicated the importance of TPX2 K429 lactylation in tumour growth in vivo.

### TPX2 lactylation is necessary for cell cycle progression by increasing AURKA phosphorylation

TPX2, a key regulator of mitosis, profoundly affects the cell cycle process (Wittmann et al, 2000; Gruss et al, 2002). To explore the potential mechanism by which TPX2 lactylation affects tumour progression, we first investigated the level of TPX2 lactylation during cell cycle using Hep3B cells synchronized in different cell cycle phases. The results showed that TPX2 lactylation was increased along with its expression during cell cycle progression (Fig S4A), suggesting that TPX2 lactylation might be necessary for its function during cell cycle. Then, we examined the effect of TPX2 lactylation on the cell cycle. The cell cycle phase of HepG2 cells with endogenous TPX2 knockdown and Flag-TPX2$^{WT}$ or Flag-TPX2$^{K249R}$ re-expression was analysed via flow cytometry. The results revealed that TPX2 knockdown led to the increased percentages of HepG2 cells at the G2/M phase, which could be restored by re-expressing TPX2$^{WT}$ but not TPX2$^{K249R}$ (Fig 4A). Similar results were observed in Hep3B cells (Fig S4B). Moreover, GSK2837808A treatment resulted in delayed cell cycle in HepG2 cells re-expressing either TPX2$^{WT}$ or TPX2$^{K249R}$ (Fig 4B). Overall, our findings indicated that TPX2 lactylation might promote tumour growth through regulating the cell cycle.

TPX2 is critical for the cell cycle regulation because it serves as an AURKA activator. By binding to autophosphorylated AURKA, TPX2 stabilizes the phosphorylation of threonine 288 (T288), thereby maintaining its kinase activity (Kufer et al, 2002; Bayliss et al, 2003; Eyers et al, 2003). Interestingly, we found that TPX2$^{WT}$, but not TPX2$^{K249R}$, could restore the decrease in AURKA T288 phosphorylation caused by TPX2 knockdown (Figs 4C and S4C). Moreover, GSK2837808A treatment resulted in similar levels of AURKA T288 phosphorylation in HepG2 cells re-expressing TPX2$^{WT}$ to that in the cells re-expressing TPX2$^{K249R}$ (Fig 4D), suggesting that TPX2 lactylation was required for maintaining AURKA T288 phosphorylation. We then employed AURKA inhibitor alisertib to treat HepG2 cells with endogenous TPX2 knockdown and Flag-tagged TPX2$^{WT}$ or TPX2$^{K249R}$ re-expression and detected AURKA T288 phosphorylation. The results showed that AURKA T288 phosphorylation in HepG2 cells treated with alisertib was reduced to a comparable extent as that in the TPX2$^{K249R}$ group treated with DMSO (Fig 4E). Further flow cytometric analysis revealed that both alisertib treatment and TPX2$^{K249R}$ re-expression led to delayed cell cycle in HepG2 cells with endogenous TPX2 knockdown (Fig 4F). Consistently, re-expression of TPX2$^{WT}$, but not TPX2$^{K249R}$, facilitated the cell growth of HepG2 cells with endogenous TPX2 knockdown, which was abolished by alisertib treatment (Figs 4G and S4D). These results further confirmed that TPX2 lactylation regulated the cell cycle in a manner dependent on the activity of AURKA.

To elucidate the underlying mechanism by which TPX2 lactylation regulates AURKA T288 phosphorylation and the cell cycle, we first investigated whether TPX2 lactylation affected the interaction between TPX2 and AURKA. However, the Co-IP assay in HepG2 and Hep3B cells overexpressing GFP-AURKA and Flag-TPX2$^{WT}$ or Flag-TPX2$^{K249R}$ revealed TPX2$^{K249R}$ had no effect on the interaction between AURKA and TPX2 (Figs 4H and S4E). In mitosis, both TPX2 and AURKA localize to the spindle, contributing to the maturation of spindle assembly (Bayliss et al, 2003; Polverino et al, 2021; Asteriti et al, 2023). Confocal fluorescence microscopy with antibody against AURKA and mCherry-tagged TPX2$^{WT}$ or TPX2$^{K249R}$ verified that either TPX2$^{WT}$ or TPX2$^{K249R}$ could colocalize with AURKA in the nuclei at the

---

knockdown using IP samples as indicated. **(C)** IP assay showing the protein interaction between CBP and TPX2 in HepG2 cells. **(D)** Western blot analysis of the lactylation of TPX2 in HepG2 cells with Flag-TPX2$^{WT}$ or Flag-TPX2$^{K249R}$ overexpression and CBP knockdown using IP samples as indicated. **(E)** Western blot analysis of the lactylation of TPX2 in HEK293T cells with Flag-TPX2 and delactylase overexpression using IP samples as indicated. Blot bands were quantified by ImageJ, and the fraction of lactylated TPX2 was represented by the ratio of the Kla band to the immunoprecipitated Flag-TPX2 band. **(F)** Western blot analysis of the lactylation of TPX2 in HepG2 cells with Flag-TPX2 and HA-HDAC1 overexpression using IP samples as indicated. **(G)** IP assay showing the protein interaction between HDAC1 and TPX2 in HepG2 cells. **(H)** Western blot analysis of the lactylation of TPX2 in HepG2 cells with Flag-TPX2$^{WT}$ or Flag-TPX2$^{K249R}$ and HA-HDAC1 overexpression using IP samples as indicated. Source data are available for this figure.

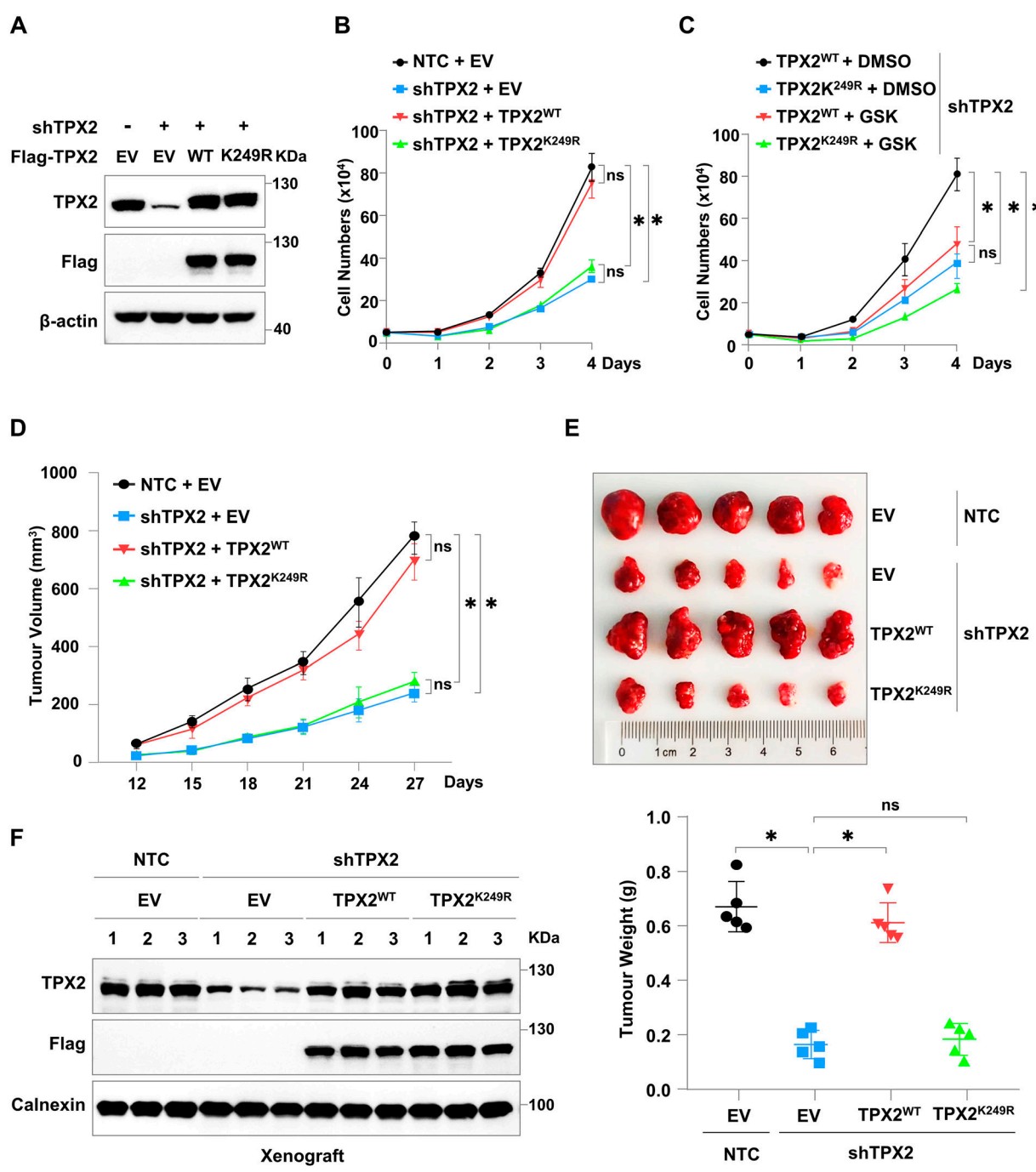

**Figure 3. TPX2 lactylation promotes tumour growth in vitro and in vivo.**
**(A, B)** Cell growth analysis of HepG2 cells with endogenous TPX2 knockdown and Flag-TPX2$^{WT}$ or Flag-TPX2$^{K249R}$ re-expression. Western blotting was used to determine the protein levels of TPX2 in HepG2 cells with the indicated genotypes. **(C)** Cell growth analysis of HepG2 cells with endogenous TPX2 knockdown and Flag-TPX2$^{WT}$ or Flag-TPX2$^{K249R}$ re-expression. HepG2 cells with indicated genotypes were treated with GSK2837808A (50 $\mu$M) for 4 d. **(D)** Tumour growth analysis of HepG2 xenografts with endogenous TPX2 knockdown and Flag-TPX2$^{WT}$ or Flag-TPX2$^{K249R}$ re-expression. The cells with indicated genotypes (5 × 10$^6$ cells per mouse) were subcutaneously injected into nude mice (n = 5 for each group). Tumour sizes were measured starting 12 d after inoculation. **(E)** Tumour weights of HepG2 xenografts with endogenous TPX2 knockdown and Flag-TPX2$^{WT}$ or Flag-TPX2$^{K249R}$ re-expression. Images of the tumours are shown at the top. **(F)** Western blot analysis of the protein levels of TPX2 in HepG2 xenograft tumours of the indicated genotypes. **(B, C, D, E)** Data information: statistical significance was determined by two-way ANOVA (B, C, D, E), ns, not significant, *$P < 0.05$.
Source data are available for this figure.

interphase and spindle microtubules at the mitotic phase (Fig S4F). These data indicated that TPX2 lactylation did not affect its cellular localization and the protein interaction between TPX2 and AURKA,

but was required for the mitosis. Protein phosphatase 1 (PP1) has been reported to dephosphorylate AURKA in the absence of TPX2 (Katayama et al, 2001; Eyers & Maller, 2004). Therefore, we examined

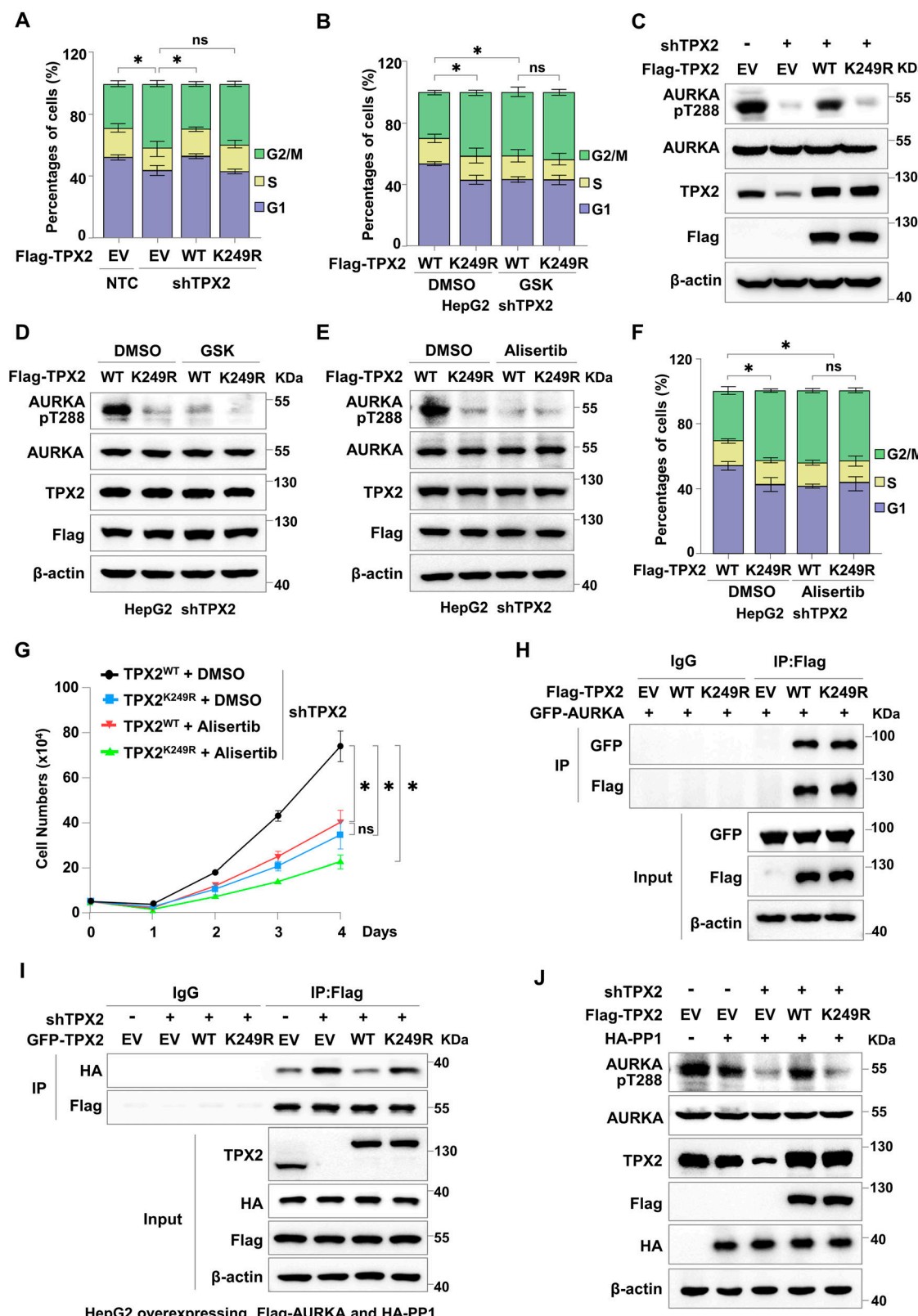

**Figure 4. TPX2 lactylation is necessary for cell cycle regulation by increasing Aurora kinase A (AURKA) phosphorylation.**
**(A)** Flow cytometric analysis of the cell cycle distribution of HepG2 cells with endogenous TPX2 knockdown and Flag-TPX2[WT] or TPX2[K249R] re-expression. The data are presented as the mean ± SEM of three independent experiments, n = 3. **(B)** Flow cytometric analysis of the cell cycle distribution of HepG2 cells with endogenous TPX2 knockdown and Flag-TPX2[WT] or Flag-TPX2[K249R] re-expression. HepG2 cells with the indicated genotypes were treated with GSK2837808A (75 μM) for 24 h. The data are

whether TPX2 lactylation regulates the interaction between AURKA and PP1. The Co-IP assay indicated that the re-expression of TPX2$^{WT}$, but not TPX2$^{K249R}$, could disrupt the strong interaction between AURKA and PP1 in HepG2 cells with endogenous TPX2 knockdown (Fig 4I). In addition, AURKA T288 phosphorylation was markedly reduced in HepG2 cells with PP1 overexpression and TPX2 knockdown, which could be restored by re-expressing TPX2$^{WT}$, but not TPX2$^{K249R}$ (Fig 4J). In conclusion, our data demonstrated that TPX2 lactylation facilitated the cell cycle progression by protecting AURKA from PP1-mediated dephosphorylation in HCC cells.

### Inhibition of the lactate/TPX2 lactylation/AURKA axis suppresses in vivo HCC tumour growth

To assess the effects of lactate/TPX2 lactylation/AURKA axis inhibition as an intervention for HCC progression in vivo, we employed xenograft model mice generated by subcutaneous inoculation with endogenous TPX2 knockdown HepG2 cells re-expressing TPX2$^{WT}$ or TPX2$^{K249R}$. When the average tumour size reached ~50–100 mm$^3$, GSK2837808A (6 mg/kg) (Gupta et al, 2021; Yan et al, 2023) or alisertib (30 mg/kg) (Hao et al, 2021; Lin et al, 2024) was intraperitoneally injected every 3 d. The re-expression of TPX2$^{WT}$, but not TPX2$^{K249R}$, facilitated the in vivo tumour growth of HepG2 xenografts with endogenous TPX2 knockdown, which was abolished by treatment with both LDHA inhibitor, GSK2837808A, and AURKA inhibitor, alisertib (Fig 5A and B). The knockdown and overexpression efficiency in xenograft was confirmed by Western blot (Fig S5). These results indicated that the regulatory effect of TPX2 lactylation on in vivo HCC tumour growth depended on lactate production and AURKA activity.

Thus, our data demonstrate that lactate-induced TPX2 lactylation is required for cell cycle regulation and HCC progression by protecting AURKA from PP1-mediated dephosphorylation to maintain its activation (Fig 5C).

# Discussion

Altered metabolism and growth signalling autonomy are the hallmarks of cancer cells (Hanahan & Weinberg, 2011). Cancer cells alter nutrient uptake and use to fulfil their need for sustained cell proliferation (Martinez-Reyes & Chandel, 2021). However, how cancer metabolism affects cellular proliferation and cell cycle

progression beyond nutrition and biosynthesis remains largely unclear. Recent work has revealed the nonmetabolic function of lactate in regulating the proliferative state. For example, accumulated lactate binds and inhibits the SUMO protease SENP1 and thus stimulates efficient mitotic exit in proliferative human cells by increasing SUMOylation-dependent remodelling of APC/C (Liu et al, 2023). In this study, via lactylome analysis of clinical HCC tumours and a series of knockdown and overexpression lines of HCC cells and YAP5SA-induced HCC mouse models, we found that lactate induced TPX2 K249 lactylation in HCC tumour tissues. Our results further indicated that the inhibition of lactate production or TPX2 lactylation led to cell cycle delay in HCC cells and suppressed in vivo tumour growth. Collectively, these studies show that lactate can directly control the cell cycle and proliferation independent of its metabolic function in cancer cells.

Several metabolites serve as signalling molecules to promote cancer progression by regulating PTMs, including acetylation, methylation, and succinylation. Dynamic acetylation, phosphorylation, ubiquitination, and methylation modulate key cell cycle regulators, which are important for maintaining genomic stability and preventing abnormal cell division (Li et al, 2021; Sun et al, 2022; Geffen et al, 2023). Therefore, the emerging role of PTMs on cell cycle regulators has garnered increasing interest as a potential strategy for cancer intervention. Lactylation is a lactate-driven PTM that has recently been reported to contribute to epigenetic regulation (Zhang et al, 2019; Wang et al, 2023), homologous recombination repair (Chen et al, 2024a), and metabolic reprogramming (Chen et al, 2024c). However, the functions of lactylation in the cell cycle remain unclear. Here, we demonstrated that TPX2 lactylation contributed to the cell cycle regulation and cancer cell proliferation by preventing PP1 from binding to AURKA to increase AURKA phosphorylation, suggesting that TPX2 lactylation is a critical mechanism for lactate accumulation contributing to cell cycle regulation.

The mitotic kinase AURKA has gained much attention as a potential therapeutic target against cancer because of its essential role in the cell cycle (Yan et al, 2016; Du et al, 2021). Although clinical trials have demonstrated the efficacy of alisertib, a selective inhibitor against AURKA, to decrease tumour growth, alisertib alone has also shown elevated levels of toxicity (Gorgun et al, 2010). Therefore, recent clinical trials have included a combination of therapies to increase its effectiveness and decrease its toxicity. TPX2 is an important AURKA interactor that is able to modulate its activity. We found that TPX2 is lactylated in HCC tumour tissues.

presented as the mean ± SEM of three independent experiments, n = 3. **(C)** Western blot analysis of AURKA T288 phosphorylation in HepG2 cells with endogenous TPX2 knockdown and Flag-TPX2$^{WT}$ or Flag-TPX2$^{K249R}$ re-expression. **(D)** Western blot analysis of AURKA T288 phosphorylation in HepG2 cells with endogenous TPX2 knockdown and Flag-TPX2$^{WT}$ or Flag-TPX2$^{K249R}$ re-expression. HepG2 cells with indicated genotypes were treated with GSK2837808A (75 $\mu$M) for 24 h. **(E)** Western blot analysis of AURKA T288 phosphorylation in HepG2 cells with endogenous TPX2 knockdown and Flag-TPX2$^{WT}$ or Flag-TPX2$^{K249R}$ re-expression. HepG2 cells with the indicated genotypes were treated with alisertib (1 $\mu$M) for 24 h. **(F)** Flow cytometric analysis of the cell cycle distribution of HepG2 cells with endogenous TPX2 knockdown and Flag-TPX2$^{WT}$ or Flag-TPX2$^{K249R}$ re-expression. HepG2 cells with the indicated genotypes were treated with alisertib (1 $\mu$M) for 24 h. The data are presented as the mean ± SEM of three independent experiments, n = 3. **(G)** Cell growth analysis of HepG2 cells with endogenous TPX2 knockdown and Flag-TPX2$^{WT}$ or Flag-TPX2$^{K249R}$ re-expression. HepG2 cells with the indicated genotypes were treated with alisertib (5 $\mu$M) for 4 d. **(H)** Co-IP assay showing the protein interaction between AURKA and TPX2$^{WT}$ or TPX2$^{K249R}$ in HepG2 cells with GFP-AURKA and Flag-TPX2$^{WT}$ or Flag-TPX2$^{K249R}$ overexpression. **(I)** Co-IP assay showing the protein interaction between AURKA and PP1. HepG2 cells with endogenous TPX2 knockdown and Flag-TPX2$^{WT}$ or Flag-TPX2$^{K249R}$ re-expression were infected with lentiviruses carrying Flag-AURKA and HA-PP1 plasmids. **(J)** Western blot analysis of AURKA T288 phosphorylation in HepG2 cells with endogenous TPX2 knockdown and Flag-TPX2$^{WT}$ or Flag-TPX2$^{K249R}$ re-expression. HepG2 cells with the indicated genotypes were infected with lentiviruses carrying HA-PP1 plasmids. **(A, B, F, G)** Data information: statistical significance was determined by two-way ANOVA (A, B, F, G), ns, not significant, *$P$ < 0.05.
Source data are available for this figure.

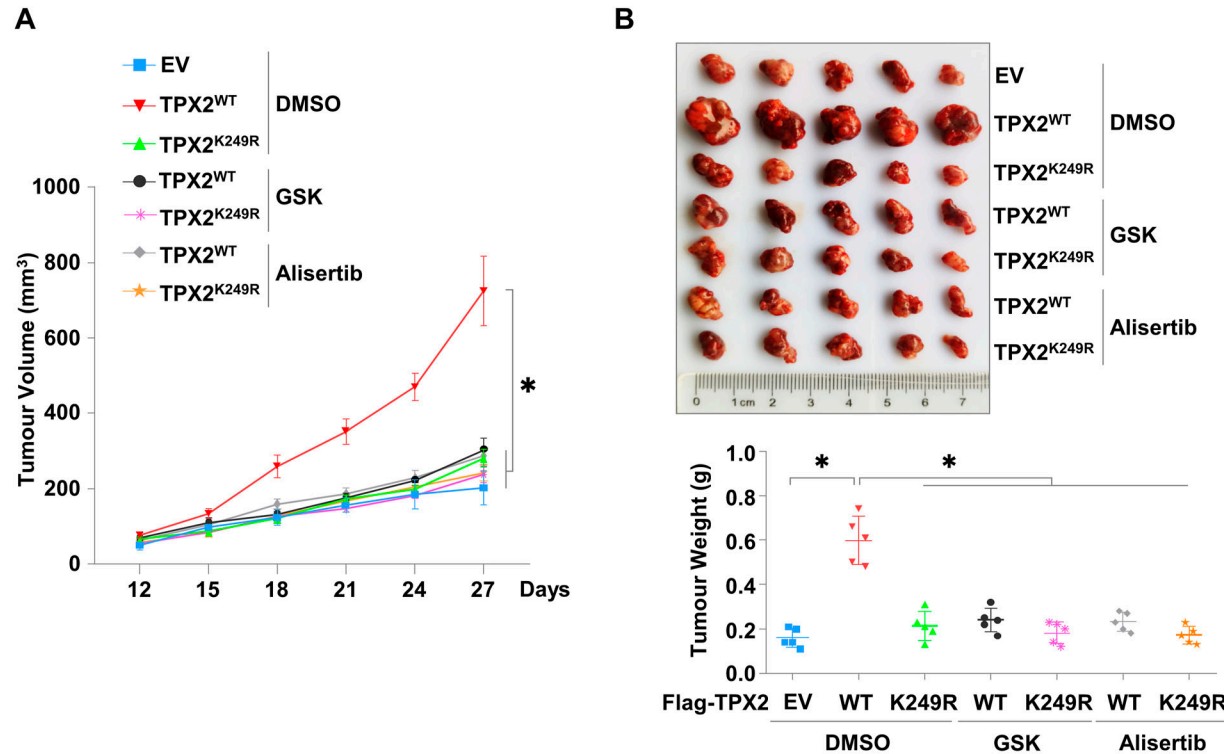

More importantly, the inhibition of lactate production via an LDHA inhibitor or TPX2 lactylation significantly suppressed in vivo HCC tumour growth, suggesting that TPX2 lactylation may serve as potential targets for new cancer interventions. Because inhibition of lactate production may have a broad effect, considering the critical role of TPX2 lactylation in cancer progression, further work will be required to generate inhibitors that could specifically suppress TPX2 K249 lactylation.

In conclusion, our results reveal that TPX2 lactylation promotes HCC progression by disrupting the interaction between AURKA and PP1 to facilitate AURKA activation and cell cycle progression, suggesting that targeting the lactate/TPX2 lactylation/AURKA axis may be a promising strategy for development of HCC interventions.

# Materials and Methods

### Cell culture

Human liver cancer cell lines (HepG2 and Hep3B) and a human renal epithelial cell line (HEK293T) were cultured in DMEM (Gibco). All the cell lines were tested for Mycoplasma contamination, and no cell lines were contaminated. All the cells were supplemented with 10% FBS (BI) and 1% penicillin–streptomycin (Invitrogen) and were cultured in a humidified incubator at 37°C and 5% $CO_2$. All drugs used in this study are listed in Table S1.

### Plasmid construction and establishment of stable cell lines

The sequences of TPX2, the TPX2 mutant (K249R), AURKA, HDAC1, CBP, and PP1 were inserted into the pSin-3×FLAG, pSin-HA, and pSin-GFP vectors (Addgene). The shRNAs against TPX2, LDHA, and CBP in the pLKO vector were purchased from a commercial supplier (Sigma-Aldrich). All shRNA targeting sequences are listed in Table S2. All shRNAs, as well as the Δ8.9 and VSVG plasmids, were transfected into PEI (Invitrogen)-supplemented HEK293T cells for 48 h, after which the supernatants of the HEK293T cells were filtered for virus collection. The cells were infected in the presence of 8 $\mu$g/ml polybrene (Sigma-Aldrich), and puromycin was used to select the infected cells.

### Western blot assay

Cultured cells were lysed in RIPA buffer (50 mM Tris–HCl [pH 8.0], 150 mM NaCl, 5 mM EDTA, 0.1% SDS, and 1% NP-40) supplemented with protease inhibitor cocktails (Roche) and 100 $\mu$M phenylmethylsulphonyl fluoride (PMSF). The protein concentration in the lysate was quantified via the Bradford method (Sangon Biotech), and the lysate was heated at 100°C for 5 min. Equal amounts of protein were loaded and separated on sodium dodecyl sulphate–polyacrylamide gel electrophoresis (SDS–PAGE) gels and transferred to NC membranes. The membranes were blocked with 5% milk in TBST buffer. Then, the membranes were incubated with primary antibodies overnight at 4°C and with secondary antibodies for 1–2 h at 37°C. Western ECL substrate (Tanon) was added, and the blots were imaged immediately on a Tanon-5300 chemiluminescence imaging system (Tanon Science and Technology). The primary antibodies used for immunoblotting included those against TPX2, AURKA, AURKA-pT288, Pan-Kla, LDHA, HDAC1, CBP (CREBBP), FLAG-Tag, HA-Tag, GFP-Tag, $\beta$-actin, and calnexin, which are summarized in Table S1. The shown blots are a representative example out of at least three performed ones.

### Immunoprecipitation

Cultured cells were lysed in IP buffer (50 mM Tris–HCl [pH 7.5], 150 mM NaCl, 5 mM EDTA, 0.1% SDS, and 1% NP-40) at 4°C for 1 h. After centrifugation, the supernatants were added to protein A/G beads (Thermo Fisher Scientific) to preclear for 1 h. After protein quantification, the corresponding antibodies were added and incubated at 4°C for 12 h. Then, the solution was incubated with protein A/G beads at 4°C for 1 h. After incubation, the beads were washed with IP buffer and boiled at 100°C for 8 min. The samples were subsequently analysed by the Western blot assay. The shown blots are a representative example out of at least three performed ones.

### In vitro lactylation assay

The Flag-tagged TPX2 proteins were incubated with HA-tagged CBP proteins, which were purified from HEK293T cells in reaction buffer (50 mM Hepes, pH 7.8, 30 mM KCl, 0.25 mM EDTA, 5.0 mM $MgCl_2$, 2.5 mM DTT) with 20 $\mu$M lactyl-CoA. Reactions were incubated at 30°C for 30 min. Next, 5 × SDS loading buffer was added to the reaction and boiled for 5 min at 100°C. Samples were separated by SDS–PAGE and immunoblotted with indicated antibodies.

### Cell growth assay

A total of 3 × 10⁴ or 5 × 10⁴ cells were seeded in 12-well plates (NEST) and cultured in a humidified incubator at 37°C and 5% $CO_2$. Then, cell counts were performed on days 1, 2, 3, and 4, and cell

---

**Figure 5. Inhibition of the lactate/TPX2 lactylation/Aurora kinase A axis suppresses in vivo hepatocellular carcinoma tumour growth.**
**(A)** Tumour growth analysis of HepG2 xenografts with endogenous TPX2 knockdown and Flag-TPX2[WT] or Flag-TPX2[K249R] re-expression subjected to the indicated treatments. The cells with the indicated genotypes (5 × 10⁶ cells per mouse) were subcutaneously injected into nude mice (n = 5 for each group). When the average tumour size reached ~50–100 mm³, GSK2837808A (6 mg/kg) or alisertib (30 mg/kg) was intraperitoneally injected every 3 d. Tumour sizes were measured starting at 12 d after inoculation. **(B)** Tumour weights of HepG2 xenografts with endogenous TPX2 knockdown and Flag-TPX2[WT] or Flag-TPX2[K249R] re-expression subjected to the indicated treatments. Images of tumours are shown at the top. **(C)** Schematic cartoon showing that lactate-induced TPX2 lactylation is required for cell cycle regulation and hepatocellular carcinoma progression by protecting Aurora kinase A from PP1-mediated dephosphorylation to maintain its activation. **(A, B)** Data information: statistical significance was determined by two-way ANOVA (A, B), ns, not significant, *P < 0.05.
Source data are available for this figure.

proliferation curves were plotted. The data represent the means ± SDs of three independent experiments.

### In vitro inhibitor assays

The cell lines were plated in 96-well tissue culture–treated plates in 2 ml of media supplemented with DMSO or chemical inhibitors for various periods of time. At the end of the incubation, cell viability was assessed via the Cell Counting Kit-8 (CCK8) assay (MCE). The results were normalized to the growth of cells in media containing an equivalent volume of DMSO. The effective concentration at which 50% inhibition of proliferation occurred (IC50) was determined via GraphPad Prism 5.0 software.

### Cell cycle analysis

The cells were harvested via trypsinization, washed twice with PBS containing 5% FBS, fixed in 70% ethanol, and then stained with 20 $\mu$g/ml propidium iodide (PI; Sigma-Aldrich) containing 20 $\mu$g/ml RNase (Fermentas). The stained cells were analysed via flow cytometry (BD Biosciences). The graph was plotted via FlowJo 7.6.5 software (FLOWJO, LLC).

### Immunofluorescence

Cells were fixed with methanol: acetone (1:1) for 20 min. Cells were washed two times using PBS and permeabilized with 0.2% Triton X-100 at RT for 15 min. Before staining, cells were blocked with 2% BSA for 1 h at RT. Next, cells were incubated with primary antibodies against AURKA (10297-1-AP; PTG), CBP (7389; CST), HDAC1 (34589T; CST), or tubulin (ab195887; Abcam) overnight at 4°C. After washing three times with PBS, cells were incubated with anti-rabbit secondary antibodies conjugated to CoraLite 594 (ab150080; Abcam) or CoraLite 555 (RGAR003; PTG) at RT for 1 h. After washing three times with PBS, cells were incubated with DAPI at RT for about 2 min. After washing, cells were mounted with anti-fade solution and visualized using a fluorescence microscope.

### Animal studies

All the animal studies were conducted with the approval of the Animal Research Ethics Committee of South China University of Technology. Five-week-old male nude mice (BALB/c) were randomly assigned to experimental groups. For xenograft experiments, equal numbers of HepG2 ($5 \times 10^6$) cells with the indicated genotype mixed with Matrigel were subcutaneously injected into nude mice (SJA Laboratory Animal Company). For GSK2837808A or alisertib treatment, when the tumours reached 50–100 mm$^3$, the mice were randomly divided into different treatment groups (n = 5). GSK2837808A or alisertib was dissolved in DMSO and diluted in PBS. GSK2837808A was intraperitoneally injected every 3 d at a dose of 6 mg/kg. Alisertib was intraperitoneally injected every 3 d at a dose of 30 mg/kg. The tumour volume was measured every 3 d after inoculation via digital callipers and calculated via the following equation: volume = length × width$^2$ × 0.52.

### Statistical analysis

All experimental data are presented as the mean ± SD or mean ± SEM from at least three independent experiments. Statistical analysis was performed via a $t$ test and two-way ANOVA to compare two groups of independent samples via GraphPad Prism 7 (GraphPad Software). Statistical significance is indicated by $P < 0.05$.

## Data Availability

The previously published lactylome data of the HCC cohort reanalysed in this study were available in the NODE database (The National Omics Data Encyclopedia) with the accession code OEP002852. All other data generated or analysed in this study and source data are available upon request.

## Supplementary Information

## Acknowledgements

This work was supported in part by grants from the National Natural Science Foundation of China (82072656, 82322050, 82341013, 82130087, 92357301).

### Author Contributions

S Liu: conceptualization, data curation, software, formal analysis, validation, investigation, and visualization.
J Cai: conceptualization, data curation, software, validation, and investigation.
X Qian: software and methodology.
J Zhang: software and investigation.
Y Zhang: software and visualization.
X Meng: software and formal analysis.
M Wang: software and visualization.
P Gao: conceptualization and resources.
X Zhong: conceptualization, resources, and methodology.

### Conflict of Interest Statement

The authors declare that they have no conflict of interest.

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
