## [Reviewer comments · Life Science Alliance]

Life Science Alliance

TPX2 lactylation is required for the cell cycle regulation and hepatocellular carcinoma progression

Shengzhi Liu, Jin Cai, Xiaoyu Qian, Junjiao Zhang, Yi Zhang, Xiang Meng, Mingjie Wang, Ping Gao, and Xiuying Zhong
DOI: <https://doi.org/10.26508/lsa.202402978>

Corresponding author(s): Xiuying Zhong, South China University of Technology and Ping Gao, University of Science and Technology of China

Review Timeline:

Submission Date:	2024-08-03
Editorial Decision:	2024-09-13
Revision Received:	2024-12-31
Editorial Decision:	2025-01-27
Revision Received:	2025-02-27
Editorial Decision:	2025-02-27
Revision Received:	2025-03-02
Accepted:	2025-03-03

Transaction Report:

September 13, 2024

Re: Life Science Alliance manuscript #LSA-2024-02978-T

Prof. Ping Gao
University of Science and Technology of China
School of Life Sciences
University of Science and Technology of China, No.96, JinZhai Road Baohe District, Hefei, Anhui, 230027, P.R. China
Hefei, Anhui 230026
China

Dear Dr. Gao,

Thank you for submitting your manuscript entitled "TPX2 lactylation is required for the cell cycle regulation and hepatocellular carcinoma progression" to Life Science Alliance. The manuscript was assessed by expert reviewers, whose comments are appended to this letter. We invite you to submit a revised manuscript addressing the Reviewer comments.

Thank you for this interesting contribution to Life Science Alliance. We are looking forward to receiving your revised manuscript.

Sincerely,

B. MANUSCRIPT ORGANIZATION AND FORMATTING:

Reviewer #1 (Comments to the Authors (Required)):

In this manuscript, Liu and colleagues explore the correlation between metabolic reprogramming and cell cycle regulation in HCC by characterizing TPX2 lactylation. This PTM is emerging as a novel modulator of biological processes and it is of interest to identify its role in control of cell cycle progression and tumorigenesis. A set of data in the paper identify TPX2 as a lactylated substrate, and the characterization of the residue and involved enzymes are provided. Then the relevance of this modification for cell cycle regulation, AurkA activation and cell transformation is addressed. Although the paper provides an interesting new layer in the control of the activity of 2 key mitotic regulators overexpressed in cancer, I find that the drawn conclusions -particularly in the second part of the paper- are not sufficiently supported by the data. Therefore in my opinion major revisions are needed prior to publication, as indicated below.

- TPX2 (as well as AurkA) is a cell cycle regulated protein and its protein levels and PTM are expected to fluctuate during the cell cycle. In this respect, and in the light of the focus of the manuscript on cell cycle regulation, it is very important: 1) to perform the experiments carried out with endogenous TPX2 in cultures synchronized in different cell cycle and mitotic stages to show whether TPX2 lactylation occurs in specific cell cycle windows; 2) to provide a cell cycle profile of the cultures where treatments, overexpression or silencing are performed (e.g., 1D,E,F; 2AB, E,...), to rule out the possibility that the observed effects are indirectly induced by a different distribution of cell cycle phases.

- authors propose that TPX2 lactylation is "induced" in HCC tumour tissues (Fig. 1I), but the levels of TPX2 itself are very different in the T and N conditions. A ratio of lactylated/total TPX2 would be more informative. Moreover, additional evaluation of TPX2 lactylation in non transformed and tumor cell lines is needed to corroborate this conclusion. This is very important since authors propose that increased lactylation (of a larger fraction of TPX2? Maintained over a longer time?) is relevant for tumor progression.

- I find that the experiments in the presence of the LDHA inhibitor GSK2837808A are difficult to interpret, because I find unlikely TPX2 to be the unique substrate, and hence it is possible that effects are indirect. Conclusions on the role of TPX2 lactylation which are based on these experiments should be mitigated.

- the cellular set up of silencing endogenous TPX2 and re-expressing WT or K249R is used as a main experimental system. Authors sometimes name this as "overexpression" and other times "re-expression". It is important to provide a quantitative characterization and to clarify if exogenous TPX2 is expressed in excess or is replacing the endogenous one at about physiological levels. Interpretation of data may be different depending on the situation.

- Is the K249R TPX2 correctly localizing at the interphase nuclei and mitotic spindle microtubules? How is AurkA localized under these conditions?

- FACS analyses would be easier to interpret if the percentages are clearly indicated. Still, the accumulation of G2/M cells following TPX2 silencing is milder than expected - given that a strong mitotic arrest is usually seen after TPX2 depletion- and is not in line with the sentence "the G2/M phase was arrested". Is it possible that the duration of the experiment is not optimal with M-arrested cells eventually dying of slipping out of mitosis? This would explain the limited G2/M increase but at the same time would complicate the interpretation of data.

- Related to the previous point, how is mitotic progression and mitotic spindle structure in WT and K249R expressing cells? Time lapse experiments and IF would provide important information to interpret the role of this PTM. PI-based FACS analysis and growth curves provide a general picture of the cell cultures that may depend on very different underlying conditions. Authors state that TPX2 lactylation contributes to the G2/M transition, but I feel that the actual contribution to the progression through G2 and mitosis is not provided at the current stage.

- I do not regard experiments with Alisertib to be conclusive. First, treatments of parental cells with no manipulation of TPX2 levels should be shown as controls. The used conditions to my knowledge are expected to have a strong effect on cell viability and tumor progression, independently on the state of TPX2. This applies to the GSK2837808A treatment (Fig 5) too. In addition, Alisertib is effective on AurkA in the nanomolar range, while treatments in Fig. 4 are made with 10 M for 24 hours. Under these conditions authors are possibly observing the effects of a strong AurkA (and maybe of other kinases) inhibition rather than the interplay between TPX2 lactylation and AurkA activation and I do not feel data provided in the paper are sufficient to conclude that the effects of TPX2 lactylation on cell cycle progression and tumorigenesis are all dependent on AurkA.

- The PP1 experiments are interesting but no possible explanation on the mechanisms is provided or tested. Why lactylation, without affecting AurkA/TPX2 binding, should affect the interaction between AurkA and PP1?

Minor:

- Figure 1C: it would be interesting to show the supernatant of the IP and state the proportion of loaded samples, to provide an estimate of the fraction of TPX2 that is lactylated
- Since lactylation is a relatively newly identified PTM, it would be important to add some controls for the experimental conditions and for the reagents, using well characterized lactylated proteins.
- Figure 2A: I expect the HA blots corresponding to different regions of the filter are shown at the same exposure, to show comparable expression. This should be stated in Figure legend.
- Fig. 3C: it is not easy to follow the asterisks labels. In particular, is the most right label referring to the WT/DMSO condition versus all other 3 conditions? If so, is not the WT/DMSO versus K249R/DMSO already shown?
- Since the results are mostly shown as western blot data, is it important to state in the Methods section that for WB and IP experiments the shown blots are a representative example out of at least 3 performed ones.

Reviewer #2 (Comments to the Authors (Required)):

In the manuscript, Liu et al suggested a possible role of TPX2 lactylation in regulating cell cycle progression and HCC tumorigenesis. Although the concept TPX2 lactylation could be interesting and valuable, the current conclusion is preliminary and requires additional robust data and evidence for validation.

There are several severe concerns that should be addressed.

1. In this study, the authors mentioned TPX2 lactylation at K249 in hepatocellular carcinoma (HCC) tumor tissues. However, this point lacks substantial supporting data such as mass spectrometry (MS) spectra, Western blotting (WB) with specific antibodies, and other data obtained from human HCC samples independent of publicly available data.
2. The authors indicated that CBP and HDAC1 regulate this process. However, the specificity of this regulation, as demonstrated in Figure 2A and 2E through overexpression of potential lactylases and delactylases, is not convincing. CBP and HDAC1 exhibit only slight differences compared to other enzymes. Therefore, more compelling experiments, such as RNA interference (RNAi) or knockout studies targeting the genes encoding these enzymes, are necessary to support this claim.
3. Most of the biochemical data were obtained from cell lines using ectopic transgenic experiments. However, there is a lack of reciprocal co-immunoprecipitation (co-IP) and in vivo colocalization studies of endogenous TPX2, CBP, and HDAC1 in these cells and HCC samples.
4. The manuscript also lacks the rigorous in vitro biochemical experiments to support the notion that TPX2 indeed functions as the substrate of CBP and HDAC1.

Point-by-point response to the comments

Editor's Decision:

Thank you for submitting your manuscript entitled "TPX2 lactylation is required for the cell cycle regulation and hepatocellular carcinoma progression" to Life Science Alliance. The manuscript was assessed by expert reviewers, whose comments are appended to this letter. We invite you to submit a revised manuscript addressing the Reviewer comments.

When submitting the revision, please include a letter addressing the reviewers' comments point by point. We hope that the comments below will prove constructive as your work progresses. Thank you for this interesting contribution to Life Science Alliance. We are looking forward to receiving your revised manuscript.

Response: We are very grateful for your kind decision allowing us to revise the manuscript. We also thank the reviewers for their insightful comments and suggestions. We believe that all the comments are critical and relevant and helped us substantially improve the study.

Over the past three months, we have performed additional experiments to address all the concerns and comments raised by our reviewers. We are now submitting a significantly improved manuscript along with our point-by-point response. In response to the reviewers' comments, we have appended in this file all the revised figures which are labeled as **Figure R1 to R15**.

Point-by-point response to the comments of the Reviewers

Reviewer #1:

Comments 1-1:

In this manuscript, Liu and colleagues explore the correlation between metabolic reprogramming and cell cycle regulation in HCC by characterizing TPX2 lactylation. This PTM is emerging as a novel modulator of biological processes and it is of interest to identify its role in control of cell cycle progression and tumorigenesis. A set of data in the paper identify TPX2 as a lactylated substrate, and the characterization of the residue and involved enzymes are provided. Then the relevance of this modification for cell cycle regulation, AurkA activation and cell transformation is addressed. Although the paper provides an interesting new layer in the control of the activity of 2 key mitotic regulators overexpressed in cancer, I find that the drawn conclusions -particularly in the second part of the paper- are not sufficiently supported by the data. Therefore, in my opinion major revisions are needed prior to publication, as indicated below.

Response: We are grateful for the reviewer's comments that well summarized the major findings and significance of our study. We also thank the reviewer for the specific insightful concerns which we will address one by one below.

Comments 1-2:

TPX2 (as well as AurkA) is a cell cycle regulated protein and its protein levels and PTM are expected to fluctuate during the cell cycle. In this respect, and in the light of the focus of the manuscript on cell cycle regulation, it is very important: 1) to perform the experiments carried out with endogenous TPX2 in cultures synchronized in different cell cycle and mitotic stages to show whether TPX2 lactylation occurs in specific cell cycle windows; 2) to provide a cell cycle profile of the cultures where treatments, overexpression or silencing are performed (e.g., 1D,E,F; 2AB, E,...), to rule out the possibility that the observed effects are indirectly induced by a different distribution of cell cycle phases.

Response: We appreciate the reviewer's insightful comments. Following the reviewer's suggestion, we analyzed TPX2 lactylation in Hep3B cells synchronized in different cell cycle and mitotic stages. The cell cycle profiles showed that Hep3B cells were synchronized in G1 phase, S phase, and G2/M phase, respectively (**Figure R1A**). As the reviewer mentioned, the protein levels of TPX2 fluctuate during the cell cycle. Our results confirmed that the expression of TPX2 appeared in the cells in G1 phase and was accumulated from S phase to G2/M phase (**Figure R1B**), which was consistent with the previous studies (Garrett *et al.*, 2002, Garrido *et al.*, 2016). Furthermore, immunoprecipitation (IP) analysis using Hep3B cells

synchronized in different cell cycle phases showed that TPX2 lactylation was increased along with the enhanced expression from G1 phase to G2/M phase (**Figure R1B**). These data indicated that the protein levels and lactylation of TPX2 did fluctuate during the cell cycle.

We further analyzed the cell cycle profile of HCC cells with lactate treatment or LDHA inhibition. Flow cytometry analysis revealed that lactate treatment had no effect on the cell cycle profile (**Figure R1C**), although the treatment increased TPX2 lactylation (**Fig 1D** in the original manuscript, for your convenience, we have appended the figure here as **Figure R1D**). The probable reason is that the lactylation level of TPX2 is sufficient to support cell cycle transition under normal conditions, and the additional lactate treatment would not significantly affect this process. Notably, both LDHA inhibitor GSK2837808A treatment and LDHA knockdown led to G2/M phase arrest and decreased TPX2 lactylation in HCC cells (**Figure R1E-H**), suggesting the important roles of intracellular lactate production in regulating TPX2 lactylation and cell cycle. We are grateful for the reviewer's concern regarding the regulation of TPX2 lactylation during cell cycle, and truly agree with that this is an interesting point. A recent study reported intracellular lactate concentrations in tumor cells increased from approximately 6 mM to 15-20 mM upon mitotic entry (Liu *et al.*, 2023), which might be responsible for the gradually elevated TPX2 lactylation from G1 phase to G2/M phase (**Figure R1B**). Taken together, these data suggested that TPX2 lactylation was at least partially regulated by the fluctuation of intracellular lactate levels during cell cycle.

Figure R1. TPX2 lactylation gradually increased from G1 to G2/M phase, which might be induced by intracellular lactate.

(A) Flow cytometry analysis of the cell cycle distribution of the indicated cells. Hep3B cells were synchronized using a double-thymidine block and then cells were collected immediately (G1 phase) or released to complete DMEM containing 10% FBS for 6 h (S phase) or 10 h (G2/M phase). Cells were collected and analyzed by flow cytometry.

(B) Western blot analysis of the lactylation of TPX2 in Hep3B cells using IP samples as indicated. Hep3B cells were synchronized using a double-thymidine block and then cells were collected immediately (G1 phase) or released to complete DMEM containing 10% FBS for 6 h (S phase) or 10 h (G2/M phase). Blot bands were quantified by ImageJ and the fraction of lactylated TPX2 were represented by the ratio of the Kla band to the immunoprecipitated TPX2 band, with normalization to β -actin.

(C) Flow cytometry analysis of the cell cycle distribution of HepG2 cells with Flag-TPX2 overexpression in the presence or absent of lactate (25 mM). The data were presented as the mean \pm SEM of three independent experiments, n=3.

(D) Western blot analysis of the lactylation of TPX2 in HepG2 cells with Flag-TPX2 overexpression using IP samples as indicated. HepG2 cells were treated with lactate (25 mM) for 24 h.

(E) Flow cytometry analysis of the cell cycle distribution of HepG2 cells with Flag-TPX2 overexpression in the presence or absent of GSK2837808A (75 μ M). The data were presented as the mean \pm SEM of three independent experiments, n=3.

(F) Western blot analysis of the lactylation of TPX2 in HepG2 cells with Flag-TPX2 overexpression using IP samples as indicated. HepG2 cells were treated with GSK2837808A (75 μ M) for 24 h, referred to as the GSK treatment.

(G) Flow cytometry analysis of the cell cycle distribution of HepG2 cells with Flag-TPX2 overexpression and LDHA knockdown. The data were presented as the mean \pm SEM of three independent experiments, n=3.

(H) Western blot analysis of the lactylation of TPX2 in HepG2 cells with Flag-TPX2 overexpression and LDHA knockdown using IP samples as indicated.

Comments 1-3:

Authors propose that TPX2 lactylation is "induced" in HCC tumour tissues (Fig. 1I), but the levels of TPX2 itself are very different in the T and N conditions. A ratio of lactylated/total TPX2 would be more informative. Moreover, additional evaluation of TPX2 lactylation in non-transformed and tumor cell lines is needed to corroborate this conclusion. This is very important since authors propose that increased lactylation (of a larger fraction of TPX2? Maintained over a longer time?) is relevant for tumor progression.

Response: We greatly appreciate the reviewer's valuable comments. Since TPX2 was expressed at substantially higher levels in HCC tumor tissues compared to that in matched non-cancerous tissues (Fig 1I in the original manuscript), we repeated the experiments using new biological replicates to determine the fraction of lactylated TPX2. The results showed that TPX2 lactylation was increased in YAP5SA-induced HCC tissues than that in adjacent noncancerous tissues (Figure R2A, see also as Fig 1I in the revised manuscript). Following the reviewer's suggestion, we further observed the lactylation levels of TPX2 in normal liver cell THLE3 and HCC cells. Consistent with the results observed in HCC tumor tissues and adjacent noncancerous tissues, the higher lactylation level of TPX2 was observed in HepG2 and Hep3B cells compared to that in THLE3 cells (Figure R2B). Warburg effect is a prominent feature in many solid tumours (Chen *et al.*, 2024a). These data demonstrated that the lactylation level of TPX2 protein was elevated in HCC cells, which might be induced by an increase in glycolytic flux to lactate in cancer cells.

Figure R2. TPX2 lactylation was induced in HCC cells and tumour tissues.

(A) Western blot analysis of the lactylation of TPX2 in adjacent noncancerous liver tissues (N) and liver cancer tumour tissues (T) from the YAP5SA-induced HCC mouse model using IP samples as indicated. Blot bands were quantified by ImageJ and the fraction of lactylated TPX2 were represented by the ratio of the immunoprecipitated TPX2 band to the corresponding Input TPX2 band, with normalization to Calnexin.

(B) Western blot analysis of the lactylation of TPX2 in the indicated cell lines using IP samples as indicated. Ponceau S staining was shown at the bottom. Blot bands were quantified by ImageJ and the fraction of lactylated TPX2 were represented by the ratio of the Kla band to the immunoprecipitated TPX2 band, with normalization to β -actin.

Comments 1-4:

I find that the experiments in the presence of the LDHA inhibitor GSK2837808A are difficult to interpret, because I find unlikely TPX2 to be the unique substrate, and hence it is possible that effects are indirect. Conclusions on the role of TPX2 lactylation which are based on these experiments should be mitigated.

Response: We thank the reviewer for the critical comments and totally agree with the opinion that TPX2 is not the unique target of the LDHA inhibitor GSK2837808A. Therefore, we firstly identified that K249 was the major lactylated site in TPX2 protein, and then investigated the effects of TPX2 lactylation on AURKA phosphorylation, cell cycle process, and tumor progression using HCC cells with endogenous TPX2 knockdown and TPX2^{WT} or TPX2^{K249R} re-expression. The results showed that re-expression of TPX2^{WT}, but not TPX2^{K249R} could largely restore the level of AURKA T288 phosphorylation, arrested cell cycle and retarded tumour growth induced by endogenous TPX2 knockdown (**Fig 3D, 4A and 4C** in the original manuscript), indicating the importance of TPX2 K429 lactylation in cell cycle regulation and tumour growth. Furthermore, we found that GSK2837808A treatment abolished the effects of TPX2^{WT} on alleviating shTPX2-induced G2/M arrest and cell growth suppression to levels similar with those expressing TPX2^{K249R} (**Fig 3C, 4B and 4D** in the original manuscript). Taken together, our data demonstrated that TPX2 lactylation was required for cell cycle regulation and cancer progression. Per our reviewer's suggestion, we discussed the effect of inhibition of lactate production via an LDHA inhibitor and TPX2 lactylation on HCC tumour growth suppression in the Discussion section in the revised manuscript.

Comments 1-5:

The cellular set up of silencing endogenous TPX2 and re-expressing WT or K249R is used as a main experimental system. Authors sometimes name this as "overexpression" and other times "re-expression". It is important to provide a quantitative characterization and to clarify if exogenous TPX2 is expressed in excess

or is replacing the endogenous one at about physiological levels. Interpretation of data may be different depending on the situation.

Response: We appreciate the reviewer for the critical comments and have improved the manuscript in this regard. We further provided a quantitative characterization to clarify the protein levels of exogenous TPX2^{WT} or TPX2^{K249R} in HCC cells expressing shTPX2 is similar to the level of the endogenous TPX2 in the NTC control cells (**Fig 3A** and **S3C** in the original manuscript; for your convenience, we have included the figures here as **Figure R3A** and **B**).

Figure R3. Re-expressing TPX2^{WT} or TPX2^{K249R} in HCC cells expressing shTPX2.

(A) Western blot analysis of the protein levels of TPX2 in HepG2 cells with the indicated genotypes. Band intensities for protein expressions in the western blot assay were quantitated by ImageJ and normalised to β -actin.

(B) Western blot analysis of the protein levels of TPX2 in Hep3B cells with the indicated genotypes. Band intensities for protein expressions in the western blot assay were quantitated by ImageJ and normalised to β -actin.

Comments 1-6:

Is the K249R TPX2 correctly localizing at the interphase nuclei and mitotic spindle microtubules? How is AurkA localized under these conditions?

Response: We thank the reviewer for the insightful comments. To address this point, we observed the localization of TPX2 and AURKA in Hep3B cells with endogenous TPX2 knockdown and TPX2^{WT} or TPX2^{K249R} re-expression. Confocal fluorescence microscopy showed that either TPX2^{WT} or TPX2^{K249R} co-localize with AURKA at the interphase nuclei and mitotic spindle microtubules (**Figure R4A** and **B**). Consistently, the IP assay indicated that TPX2^{K249R} had no effect on the interaction between AURKA and TPX2 (**Fig 4H** in the original manuscript). Notably, in the cells with endogenous TPX2 knockdown and

TPX2^{K249R} re-expression, mitotic spindles were improperly assembled, and chromosomes failed to align at the equatorial plate (**Figure R4B**, see also as **Fig S4E** in the revised manuscript). These data suggested that TPX2 lactylation did not affect the subcellular localization of AURKA or TPX2 itself, but contributed to spindle organization in mitosis, which might be responsible for the G2/M arrest in HCC cells with reduced TPX2 lactylation (**Fig 4A** in the original manuscript). A recent study reported that AURKA nuclear localization is not influenced by AURKA kinase activity, but is promoted by the co-overexpression of TPX2 in cancer cells (Asteriti *et al.*, 2023). Our results demonstrated that TPX2 lactylation promotes hepatocellular carcinoma progression by enhancing AURKA phosphorylation and activity. Taken together, these studies revealed the diverse roles of TPX2 in cancer progression, suggesting TPX2 protein and its post translational modification can serve as potential targets for cancer interventions.

Figure R4. TPX2 lactylation contributed to spindle organization in mitosis.

(A) Representative immunofluorescence staining for TPX2 and AURKA at the interphase in Hep3B cells with endogenous TPX2 knockdown and mCherry-TPX2^{WT} or mCherry-TPX2^{K249R} re-expression. The nucleus was stained with DAPI. Scare bar, 10 μ m.

(B) Representative immunofluorescence staining for TPX2 and AURKA at the mitotic phase in Hep3B cells with endogenous TPX2 knockdown and mCherry-TPX2^{WT} or mCherry-TPX2^{K249R} re-expression. The nucleus was stained with DAPI. Scare bar, 10 μ m.

Comments 1-7:

FACS analyses would be easier to interpret if the percentages are clearly indicated. Still, the accumulation of G2/M cells following TPX2 silencing is milder than expected - given that a strong mitotic arrest is usually seen after TPX2 depletion- and is not in line with the sentence "the G2/M phase was arrested". Is it possible that the duration of the experiment is not optimal with M-arrested cells eventually dying or slipping out of mitosis? This would explain the limited G2/M increase but at the same time would complicate the interpretation of data.

Response: We are very grateful to the reviewers for the valuable advice. Accordingly, we investigated the cell cycle profile of HepG2 cells expressing shTPX2s (**Figure R5A**) and showed the cell percentages in different cell cycle phases separately. The results showed that TPX2 knockdown significantly increased the percentages of cells in G2/M phase (**Figure R5B**), which was similar to the observation in liver cancer cells from the previous study (Hsu *et al.*, 2017). Recent studies revealed various functions of TPX2 involving in mitotic spindle assembly, homologous recombination repair and macrophage polarization under different contexts (Wang *et al.*, 2024, Xiao *et al.*, 2024). As the reviewer has pointed out, TPX2 silencing may lead to G2/M arrest and cell death. Nevertheless, in current study we found that the G2/M phase was arrested in the cell cycle of HCC cells with TPX2 knockdown, which could be restarted by restoring the expression of TPX2^{WT} but not TPX2^{K249R} (**Fig 4A** and **S4A** in the original manuscript), indicating that TPX2 lactylation might promote tumour growth through regulating the cell cycle.

Figure R5. TPX2 knockdown led to G2/M arrest in HCC cells.

(A) Western blot analysis of the protein levels of TPX2 in HepG2 cells with TPX2 knockdown.

(B) Flow cytometry analysis of the cell cycle distribution of HepG2 cells with TPX2 knockdown. The data were presented as the mean \pm SEM of three independent experiments, n=3.

Comments 1-8:

Related to the previous point, how is mitotic progression and mitotic spindle structure in WT and K249R expressing cells? Time lapse experiments and IF would provide important information to interpret the role of this PTM. PI-based FACS analysis and growth curves provide a general picture of the cell cultures that may depend on very different underlying conditions. Authors state that TPX2 lactylation contributes to the G2/M transition, but I feel that the actual contribution to the progression through G2 and mitosis is not provided at the current stage.

Response: We are grateful for the reviewer's insightful comments. Following the reviewer's suggestion, we observed the mitotic spindle structure in Hep3B cells with endogenous TPX2 knockdown and TPX2^{WT} or TPX2^{K249R} re-expression. Confocal fluorescence microscopy with antibodies against AURKA and Tubulin or mCherry-tagged TPX2^{WT} or TPX2^{K249R} verified the colocalization of TPX2^{WT} or TPX2^{K249R} and AURKA at the interphase nuclei and mitotic spindle microtubules (**Figure R4A** and **B**). Importantly, the cells with endogenous TPX2 knockdown and TPX2^{K249R} re-expression exhibited a defect in mitotic spindle assembly in mitotic phase (**Figure R4B**, see also as **Fig S4E** in the revised manuscript), suggesting that TPX2 lactylation is required for spindle organization in mitosis. We appreciate our reviewer for the constructive advice that helps us provide new evidence to demonstrate that TPX2 lactylation contributes to the G2/M transition.

Comments 1-9:

I do not regard experiments with Alisertib to be conclusive. First, treatments of parental cells with no manipulation of TPX2 levels should be shown as controls. The used conditions to my knowledge are expected to have a strong effect on cell viability and tumor progression, independently on the state of TPX2. This applies to the GSK2837808A treatment (Fig 5) too. In addition, Alisertib is effective on Aurka in the nanomolar range, while treatments in Fig. 4 are made with 10 μ M for 24 hours. Under these conditions authors are possibly observing the effects of a strong Aurka (and maybe of other kinases) inhibition rather than the interplay between TPX2 lactylation and Aurka activation and I do not feel data provided in the paper are sufficient to conclude that the effects of TPX2 lactylation on cell cycle progression and tumorigenesis are all dependent on Aurka.

Response: We thank the reviewer for the critical comments. Following the reviewer's suggestion, we investigated the effect of TPX2 lactylation on AURKA phosphorylation and the cell cycle with treatments of parental cells as controls. Western blot analysis showed that GSK2837808A treatment obviously decreased AURKA T288 phosphorylation in HepG2 cells compared to that in control group (**Figure R6A**). Moreover, GSK2837808A treatment resulted in similar levels of AURKA T288 phosphorylation in endogenous TPX2 knocked down HepG2 cells re-expressing TPX2^{WT} to that in the cells re-expressing TPX2^{K249R}. Further flow cytometric analysis revealed that GSK2837808A treatment also resulted in cell cycle arrest in both HepG2 cells re-expressing TPX2^{WT} or TPX2^{K249R} (**Figure R6B**). For testing GSK2837808A *in vivo*, mice were treated as described before (Gupta *et al.*, 2021, Yan *et al.*, 2023). The results showed that re-expression of TPX2^{WT}, but not TPX2^{K249R}, facilitated the *in vivo* tumour growth of HepG2 xenografts with endogenous TPX2 knockdown, which was abolished by treatment with LDHA inhibitor GSK2837808A (**Fig 5A, B** in the original manuscript). Lactate, which is derived from pyruvate in a reaction catalyzed by LDH, has been known to play key roles in various cellular processes, including tumor cell proliferation, metastatic dissemination, and immune suppression (Liberti *et al.*, 2016). Therefore, we completely agree with the reviewer that LDHA inhibitor GSK2837808A treatment has the effect on cell viability and tumor progression independently on the state of TPX2. The current study is focused on revealing the impact of lactate-induced TPX2 lactylation on cell cycle progression and tumorigenesis. Taken together, these data suggested that interventions of lactate accumulation and TPX2 lactylation may be a promising approach to mediate the cell cycle arrest in cancer cells.

We truly appreciate the reviewer for the concern regarding the dose of Alisertib. We performed additional experiments using 1 μ M Alisertib to treat HepG2 cells for 24 hours according to IC50 value of Alisertib (**Fig S4C** in the original manuscript). Flow cytometric analysis confirmed that 1 μ M Alisertib did not induce cell death in parental HepG2 cells (**Figure R6C**). Then we detected AURKA T288 phosphorylation in HepG2 cells with endogenous TPX2 knockdown and Flag-tagged TPX2^{WT} or TPX2^{K249R} re-expression in the presence or absent of Alisertib. Western blot analysis indicated that Alisertib treatment reduced the levels of AURKA T288 phosphorylation in HepG2 cells re-expressing Flag-tagged TPX2^{WT} to a comparable extent as that in the TPX2^{K249R} group treated with DMSO (**Figure R6D**). Further flow cytometric analysis revealed that both Alisertib treatment and TPX2^{K249R} re-expression led to G2/M phase arrest in HepG2 cells with endogenous TPX2 knockdown (**Figure R6E**). Considering the critical role of AURKA in mitosis and cancer progression (Dauch *et al.*, 2016), we believe that the increased AURKA T288 phosphorylation level mediated by TPX2 lactylation might not be the only reason, but must play an important role in regulating cell cycle progression and tumorigenesis.

Figure R6. TPX2 lactylation regulated the cell cycle dependent on the activity of AURKA.

(A) Western blot analysis of AURKA T288 phosphorylation in HepG2 cells with endogenous TPX2 knockdown and Flag-TPX2^{WT} or Flag-TPX2^{K249R} re-expression. HepG2 cells with indicated genotypes were treated with GSK2837808A (75 μ M) for 24 h.

(B) Flow cytometry analysis of the cell cycle distribution of HepG2 cells with endogenous TPX2 knockdown and Flag-TPX2^{WT} or Flag-TPX2^{K249R} re-expression. HepG2 cells with the indicated genotypes were treated with GSK2837808A (75 μ M) for 24 h. The data were presented as the mean \pm SEM of three independent experiments, n=3.

(C) Cell death analysis of HepG2 cells treated with 1 μ M or 10 μ M Alisertib for 24h.

(D) Western blot analysis of AURKA T288 phosphorylation in HepG2 cells with endogenous TPX2 knockdown and Flag-TPX2^{WT} or Flag-TPX2^{K249R} re-expression. HepG2 cells with the indicated genotypes were treated with Alisertib (1 μ M) for 24 h.

(E) Flow cytometry analysis of the cell cycle distribution of HepG2 cells with endogenous TPX2 knockdown and Flag-TPX2^{WT} or Flag-TPX2^{K249R} re-expression. HepG2 cells with the indicated genotypes were treated with Alisertib (1 μ M) for 24 h. The data were presented as the mean \pm SEM of three independent experiments, n=3.

Comments 1-10:

The PP1 experiments are interesting but no possible explanation on the mechanisms is provided or tested. Why lactylation, without affecting Aurka/TPX2 binding, should affect the interaction between Aurka and PP1?

Response: We appreciate the reviewer for the insightful comments. A study by Bayliss et al. reported that N-terminal 1-43 amino acids of TPX2 are sufficient for its interaction with AURKA (Bayliss *et al.*, 2003). The lactylome of the HCC cohort (Yang *et al.*, 2023) and our data (**Fig 1G** in the original manuscript) indicated that Lysine 249 in TPX2 was its major lactylation site. Further IP assay revealed that Lysine 249 of TPX2 is not necessary for the protein interaction between TPX2 and AURKA (**Fig 4H** in the original manuscript), which might be responsible for that TPX2 K429 lactylation did not affect AURKA/TPX2 binding. We agree completely with our reviewer that this is an interesting point and have tried to predict the protein structure of K429-lactylated TPX2 and the protein interaction between AURKA and PP1 in presence of TPX2^{WT} or TPX2^{K249R} using AlphaFold2. But we failed to perform the Molecular docking prediction, because the overall structure of TPX2 has not been determined (Bayliss *et al.*, 2003). Considering the critical roles of TPX2 and AURKA in regulating mitosis (Kufer *et al.*, 2002, Fu *et al.*, 2015, Poverino *et al.*, 2021), we believe that the broad functions of TPX2 lactylation and the underlying mechanisms warrant further independent study.

Comments 1-11:

Figure 1C: it would be interesting to show the Supernatant of the IP and state the proportion of loaded samples, to provide an estimate of the fraction of TPX2 that is lactylated.

Response: Thank you very much for the helpful advice. Following the reviewer's suggestion, we performed Immunoprecipitation (IP) with antibodies targeting lysine-lactylation (Kla) in HepG2 cells and detected protein levels of TPX2 in the immunoprecipitant and supernatant of the IP. The following western blot analysis confirmed that TPX2 was lactylated in HepG2 cells (**Figure R7A**). Since TPX2 protein was rarely detected in the supernatant of the IP, the proportion of lactylated TPX2 was estimated by the ratio of the TPX2 band in the IP fraction to the corresponding Input TPX2 band, with normalization to β -actin. The results showed that the fraction of lactylated TPX2 was 77% (**Figure R7A**), suggesting that a larger fraction of TPX2 was lactylated in HCC cells.

Figure R7. TPX2 was lactylated in HCC cells

(A) Western blot analysis of the lactylation of TPX2 in HepG2 cells using IP samples as indicated. Blot bands were quantified by ImageJ and the fraction of lactylated TPX2 were represented by the ratio of the immunoprecipitated TPX2 band to the corresponding Input TPX2 band, with normalization to β -actin.

Comments 1-12:

Since lactylation is a relatively newly identified PTM, it would be important to add some controls for the experimental conditions and for the reagents, using well characterized lactylated proteins.

Response: We appreciate the reviewer's valuable advice. Histone H3 Lys18 (H3K18) lactylation has been reported to play important roles in M1 macrophage polarization (Zhang *et al.*, 2019). Following the reviewer's suggestion, we detected the lactylation levels of H3K18 in HEK293T cells treated with lactate or LDHA inhibitors. Western blot analysis with anti-Lactyl-Histone H3 (Lys18) antibody showed that lactate treatment increased the lactylation of H3K18, whereas LDHA inhibitors GSK2837808A and sodium oxamate

decreased H3K18 lactylation (**Figure R8A**). Similar results were observed in TPX2 protein, suggesting that intracellular lactate induces TPX2 lactylation.

Figure R8. Intracellular lactate induced lactylation of both TPX2 and H3K18.

(A) Western blot analysis of the lactylation of TPX2 in HEK293T cells with Flag-TPX2 overexpression using IP samples as indicated. HEK293T cells were treated with lactate (25 mM), oxamate (20 mM) or GSK2837808A (75 μ M) for 24 h. Blot bands were quantified by ImageJ and the fraction of lactylated TPX2 were represented by the ratio of the Klal band to the immunoprecipitated Flag-TPX2 band, with normalization to β -actin. H3K18 lactylation was detected by western blot analysis with anti-Lactyl-Histone H3 (Lys18) antibody.

Comments 1-13:

Figure 2A: I expect the HA blots corresponding to different regions of the filter are shown at the same exposure, to show comparable expression. This should be stated in Figure legend.

Response: We appreciate the reviewer for pointing this out and apologize for overlooking this point. Actually, we compared the levels of TPX2 lactylation in HEK293T cells overexpressing the lactylases to that in the empty vector (EV) control group and performed the HA blots exposure separately. We appended western blot data of another two independent experiments here, in which HA blots corresponding to different lactylases are shown at the same exposure. Consistent with the previous experiment, the results showed that compared to EV group, the level of lactylated TPX2 was obviously increased in CBP overexpressing cells

(Figure R9A). We have replaced Fig 2A in the revised manuscript and have stated that the HA blots corresponding to different regions of the filter are shown at the same exposure in the figure legend.

Figure R9. TPX2 is lactylated by CBP.

(A) Western blot analysis of the lactylation of TPX2 in HEK293T cells with Flag-TPX2 and lactylase overexpression using IP samples as indicated. HA blots corresponding to different lactylases are shown at the same exposure. Blot bands were quantified by ImageJ and the fraction of lactylated TPX2 were represented by the ratio of the Kla band to the immunoprecipitated Flag-TPX2 band, with normalization to β -actin.

Comments 1-14:

Fig. 3C: it is not easy to follow the asterisks labels. In particular, is the rightest label referring to the WT/DMSO condition versus all other 3 conditions? If so, is not the WT/DMSO versus K249R/DMSO already shown?

Response: Thank you very much for pointing this out. We have improved the asterisks labels in Figure 3C in the revised manuscript. For your convenience, we appended the figure here as **Figure R10A**. The results showed that TPX2^{WT}, but not TPX2^{K249R}, restored the retarded cell growth induced by TPX2 knockdown in HepG2 cells, and the LDHA inhibitor GSK2837808A treatment attenuated the cell growth advantage of

HepG2 cells re-expressing TPX2^{WT} (**Figure R10A**, see also as **Fig 3C** in the revised manuscript), indicating that TPX2 lactylation was crucial for HCC cell growth.

Figure R10. Interventions of lactate accumulation and TPX2 lactylation suppressed HCC cell growth.

(A) Cell growth analysis of HepG2 cells with endogenous TPX2 knockdown and Flag-TPX2^{WT} or Flag-TPX2^{K249R} re-expression. HepG2 cells with indicated genotypes were treated with GSK2837808A (50 μ M) for 4 days.

Comments 1-15:

Since the results are mostly shown as western blot data, is it important to state in the Methods section that for WB and IP experiments the shown blots are a representative example out of at least 3 performed ones.

Response: We appreciated the reviewer for the important comments and have stated in the Methods section in the revised manuscript that the shown blots are a representative example out of at least 3 performed ones for WB and IP experiments.

Reviewer #2:

Comments 2-1:

In the manuscript, Liu et al suggested a possible role of TPX2 lactylation in regulating cell cycle progression and HCC tumorigenesis. Although the concept TPX2 lactylation could be interesting and valuable, the current conclusion is preliminary and requires additional robust data and evidence for validation. There are several severe concerns that should be addressed.

Response: We appreciate the reviewer for the positive opinion of our work and the insightful comments that help us substantially improve the current study.

Comments 2-2:

In this study, the authors mentioned TPX2 lactylation at K249 in hepatocellular carcinoma (HCC) tumor tissues. However, this point lacks substantial supporting data such as mass spectrometry (MS) spectra, Western blotting (WB) with specific antibodies, and other data obtained from human HCC samples independent of publicly available data.

Response: We truly appreciate the reviewer for the critical comments. As our reviewer mentioned, we screened the publicly lactylome of the clinical HCC cohort and found that TPX2 was lactylated at K249. As a matter of fact, we also expected to detect the levels of TPX2 lactylation in clinical HCC tumour tissues by western blot analysis using specific antibody targeting TPX2 K249 lactylation. But no commercialized antibody is available, since the previously unrecognized role for TPX2 lactylation in orchestrating cancer progression has just been noticed in current study. We also tried to generate an antibody against TPX2 K249 lactylation, but the lysine residues near lysine 249 in TPX2 protein making it difficult to specifically targeting K249 lactylation (**Figure R11A**). Instead, we had to employ a YAP5SA-induced spontaneous HCC mouse model to investigate whether TPX2 is lactylated *in vivo*. Immunoprecipitation with Pan-Kla antibody followed by western blot analysis showed that TPX2 lactylation was obviously increased in YAP5SA-induced HCC tissues compared to that in adjacent noncancerous tissues (**Fig 11** in the original manuscript). Collectively, our IP results and the data from publicly lactylome of the clinical HCC cohort demonstrated that TPX2 lactylation was induced in HCC tumour tissues.

A
Homo sapiens **KKNEEFKKLA LAGIGQPVKK SVSQVTKSVD FHFRTDERIK**
230 249 269

Figure R11. Amino acid sequence of human TPX2 from lysine 230 to lysine 269 (NP_036244.2).

Comments 2-3:

The authors indicated that CBP and HDAC1 regulate this process. However, the specificity of this regulation, as demonstrated in Figure 2A and 2E through overexpression of potential lactylases and delactylases, is not convincing. CBP and HDAC1 exhibit only slight differences compared to other enzymes. Therefore, more compelling experiments, such as RNA interference (RNAi) or knockout studies targeting the genes encoding these enzymes, are necessary to support this claim.

Response: We are grateful to the reviewers for the valuable advice. To address our reviewer's concern, we provided a quantitative characterization to assess the lactylation levels of TPX2 in HEK293T cells overexpressing the potential lactylases and delactylases. The results showed that compared to the empty vector control group, the level of lactylated TPX2 was obviously increased in CBP overexpressing cells (**Figure R12A**, see also as **Fig 2A** in the revised manuscript), and was decreased in HDAC1 overexpressing cells (**Figure R12B**, see also as **Fig 2E** in the revised manuscript).

In addition, following the reviewer's suggestion, we further screened the major regulators of TPX2 lactylation in HEK293T cells expressing shRNAs targeting the genes encoding these enzymes. IP assay indicated that the lactylation of TPX2 was markedly decreased with CBP knockdown (**Figure R12C**), whereas it was increased in cells with HDAC1 knockdown (**Figure R12D**), which was consistent with the above results. These results further verified that CBP and HDAC1 were the major regulators of TPX2 lactylation.

Figure R12. TPX2 was lactylated by CBP and delactylated by HDAC1.

(A) Western blot analysis of the lactylation of TPX2 in HEK293T cells with Flag-TPX2 and lactylase overexpression using IP samples as indicated. Blot bands were quantified by ImageJ and the fraction of lactylated TPX2 were represented by the ratio of the Kla band to the immunoprecipitated Flag-TPX2 band, with normalization to β -actin.

(B) Western blot analysis of the lactylation of TPX2 in HEK293T cells with Flag-TPX2 and delactylase overexpression using IP samples as indicated. Blot bands were quantified by ImageJ and the fraction of lactylated TPX2 were represented by the ratio of the Kla band to the immunoprecipitated Flag-TPX2 band, with normalization to β -actin.

(C) Western blot analysis of the lactylation of TPX2 in HEK293T cells with Flag-TPX2 overexpression and lactylase knockdown using IP samples as indicated. Blot bands were quantified by ImageJ and the fraction of lactylated TPX2 were represented by the ratio of the Kla band to the immunoprecipitated Flag-TPX2 band, with normalization to β -actin. The knockdown efficiency was verified by qRT-PCR analysis.

(D) Western blot analysis of the lactylation of TPX2 in HEK293T cells with Flag-TPX2 overexpression and delactylase knockdown using IP samples as indicated. Blot bands were quantified by ImageJ and the fraction of lactylated TPX2 were represented by the ratio of the Kla band to the immunoprecipitated Flag-TPX2 band, with normalization to β -actin. The knockdown efficiency was verified by qRT-PCR analysis.

Comments 2-4:

Most of the biochemical data were obtained from cell lines using ectopic transgenic experiments. However, there is a lack of reciprocal co-immunoprecipitation (co-IP) and *in vivo* colocalization studies of endogenous TPX2, CBP, and HDAC1 in these cells and HCC samples.

Response: We are very grateful to the reviewer for the constructive advice. Immunoprecipitation (IP) with antibody targeting TPX2 showed that TPX2 interacted with CBP or HDAC1 in HepG2 cells (**Fig 2C and G** in the original manuscript; for your convenience, we append the figures here as **Figure R13A**). We then performed additional IP assays using antibodies targeting CBP or HDAC1 in HepG2 cells. The results further validated the protein interaction between TPX2 and CBP or HDAC1 (**Figure R13B**, see also as **Fig S2B and S2E** in the revised manuscript).

Figure R13. TPX2 interacted with CBP or HDAC1 in HCC cells.

(A) IP assay showing the protein interaction between TPX2 and CBP or HDAC1 in HepG2 cells.

(B) IP assay showing the protein interaction between CBP or HDAC1 and TPX2 in HepG2 cells.

Following the reviewer's suggestion, we performed immunofluorescence analysis to investigate the *in vivo* co-localization of TPX2 and CBP or HDAC1. Confocal fluorescence microscopy with GFP-TPX2 and antibody against CBP showed that TPX2 co-localized with CBP in the nucleus in cancer cells (**Figure R14A**,

see also as **Fig S2D** in the revised manuscript). Similar results were observed in the tumour tissues from the YAP5SA-induced HCC mouse model (**Figure R14B**). We also verified the co-localization of TPX2 and HDAC1 in HCC cells (**Figure R14C**, see also as **Fig S2G** in the revised manuscript) and the tumour tissues from the YAP5SA-induced HCC mouse model (**Figure R14D**) by immunofluorescence analysis. Collectively, IP assays using antibodies targeting endogenous proteins in conjunction with immunofluorescence microscopy verified the association between TPX2 and CBP or HDAC1 in HCC cells.

Figure R14. TPX2 co-localized with CBP or HDAC1 in HCC cells and tumour tissue.

(A) Representative immunofluorescence staining for TPX2 and CBP in Hep3B cells. The nucleus was stained with DAPI. Co-localization analysis of immunofluorescence images was quantitated by the colocalization plugin. Scare bar, 10 μ m.

(B) Representative immunofluorescence staining for TPX2 and CBP in tumour tissues from the YAP5SA-induced HCC mouse model. The nucleus was stained with DAPI. Co-localization analysis of immunofluorescence images was quantitated by the colocalization plugin. Scare bar, 20 μ m.

(C) Representative immunofluorescence staining for TPX2 and HDAC1 in Hep3B cells. The nucleus was stained with DAPI. Co-localization analysis of immunofluorescence images was quantitated by the colocalization plugin. Scare bar, 10 μm .

(D) Representative immunofluorescence staining for TPX2 and HDAC1 in tumour tissues from the YAP5SA-induced HCC mouse model. The nucleus was stained with DAPI. Co-localization analysis of immunofluorescence images was quantitated by the colocalization plugin. Scare bar, 20 μm .

Comments 2-5:

The manuscript also lacks the rigorous *in vitro* biochemical experiments to support the notion that TPX2 indeed functions as the substrate of CBP and HDAC1.

Response: We appreciate the reviewer's valuable advice. Following the reviewer's suggestion, we performed *in vitro* biochemical assays to validate that TPX2 was lactylated by CBP based on the method from previous study (Chen *et al.*, 2024b). The *in vitro* lactylation assays using Flag-tagged TPX2 and HA-tagged CBP proteins which are purified from HEK293T cells showed that TPX2 was lactylated only in the presence of both CBP and lactyl-CoA (**Figure R15A**, see also as **Fig S2A** in the revised manuscript), suggesting that CBP mediated TPX2 lactylation in a lactyl-CoA dependent manner. Since there is no available method for *in vitro* delactylation assay, we would like to request not to perform the suggested important *in vitro* biochemical experiments to investigate that TPX2 functions as the substrate of HDAC1. It is really hard for us to establish an available method and accomplish the *in vitro* delactylation assay during the time frame allowed for this revision. We detected the levels of TPX2 lactylation in HEK293T cells with multiple deacetylases knockdown and found that HDAC1 knockdown obviously increased TPX2 lactylation (**Figure R12B**). The additional IP assays and IF co-localization assays also verified the interaction and co-localization of TPX2 and HDAC1 in HCC cells (**Figure R13** and **Figure R14**). Taken together, these results indicated that HDAC1 is the major regulator of TPX2 delactylation.

Figure R15. CBP mediated TPX2 lactylation *in vitro*.

(A) *In vitro* TPX2 lactylation assay. The Flag-TPX2 proteins were incubated with HA-CBP proteins which were purified from HEK293T cells in the presence or absent of lactyl-CoA. TPX2 lactylation was detected by western blot using anti-Pan-Kla antibody.

References

- Garrett S, Auer K, Compton DA, Kapoor TM (2002) hTPX2 is required for normal spindle morphology and centrosome integrity during vertebrate cell division. *Curr Biol* 12: 2055-9
- Garrido G, Vernos I (2016) Non-centrosomal TPX2-Dependent Regulation of the Aurora A Kinase: Functional Implications for Healthy and Pathological Cell Division. *Front Oncol* 6: 88
- Liu W, Wang Y, Bozi LHM, Fischer PD, Jedrychowski MP, Xiao H, Wu T, Darabedian N, He X, Mills EL, Burger N, Shin S, Reddy A, Sprenger HG, Tran N, Winther S, Hinshaw SM, Shen J, Seo HS, Song K et al. (2023) Lactate regulates cell cycle by remodelling the anaphase promoting complex. *Nature* 616: 790-797
- Chen H, Li Y, Li H, Chen X, Fu H, Mao D, Chen W, Lan L, Wang C, Hu K, Li J, Zhu C, Evans I, Cheung E, Lu D, He Y, Behrens A, Yin D, Zhang C (2024a) NBS1 lactylation is required for efficient DNA repair and chemotherapy resistance. *Nature* 631: 663-669
- Asteriti IA, Polverino F, Stagni V, Sterbini V, Ascanelli C, Naso FD, Mastrangelo A, Rosa A, Paiardini A, Lindon C, Guarguaglini G (2023) AurkA nuclear localization is promoted by TPX2 and counteracted by protein degradation. *Life Science Alliance* 6: e202201726
- Hsu CW, Chen YC, Su HH, Huang GJ, Shu CW, Wu TT, Pan HW (2017) Targeting TPX2 Suppresses the Tumorigenesis of Hepatocellular Carcinoma Cells Resulting in Arrested Mitotic Phase Progression and Increased Genomic Instability. *J Cancer* 8: 1378-1394
- Wang L, Zhang H, Li Y, Li L (2024) TPX2 influences the regulation of macrophage polarization via the NF-kappaB pathway in lung adenocarcinoma. *Life Sci* 340: 122437
- Xiao M, Tang R, Pan H, Yang J, Tong X, Xu H, Guo Y, Lei Y, Wu D, Lei Y, Han Y, Ma Z, Wang W, Xu J, Yu X, Shi S (2024) TPX2 serves as a novel target for expanding the utility of PARPi in pancreatic cancer through conferring synthetic lethality. *Gut*
- Gupta VK, Sharma NS, Durden B, Garrido VT, Kesh K, Edwards D, Wang D, Myer C, Mateo-Victoriano B, Kollala SS, Ban Y, Gao Z, Bhattacharya SK, Saluja A, Singh PK, Banerjee S (2021) Hypoxia-Driven Oncometabolite L-2HG Maintains Stemness-Differentiation Balance and Facilitates Immune Evasion in Pancreatic Cancer. *Cancer Res* 81: 4001-4013
- Yan J, Li W, Tian H, Li B, Yu X, Wang G, Sang W, Dai Y (2023) Metal-Phenolic Nanomedicines Regulate T-Cell Antitumor Function for Sono-Metabolic Cancer Therapy. *ACS Nano* 17: 14667-14677
- Liberti MV, Locasale JW (2016) The Warburg Effect: How Does it Benefit Cancer Cells? *Trends Biochem Sci* 41: 211-218
- Dauch D, Rudalska R, Cossa G, Nault JC, Kang TW, Wuestefeld T, Hohmeyer A, Imbeaud S, Yevsa T, Hoenicke L, Pantsar T, Bozko P, Malek NP, Longerich T, Laufer S, Poso A, Zucman-Rossi J, Eilers M, Zender L (2016) A MYC-aurora kinase A protein complex represents an actionable drug target in p53-altered liver cancer. *Nat Med* 22: 744-53
- Bayliss R, Sardon T, Vernos I, Conti E (2003) Structural basis of Aurora-A activation by TPX2 at the mitotic spindle. *Mol Cell* 12: 851-62
- Yang Z, Yan C, Ma J, Peng P, Ren X, Cai S, Shen X, Wu Y, Zhang S, Wang X, Qiu S, Zhou J, Fan J, Huang H,

- Gao Q (2023) Lactylome analysis suggests lactylation-dependent mechanisms of metabolic adaptation in hepatocellular carcinoma. *Nat Metab* 5: 61-79
- Kufer TA, Sillje HH, Korner R, Gruss OJ, Meraldi P, Nigg EA (2002) Human TPX2 is required for targeting Aurora-A kinase to the spindle. *J Cell Biol* 158: 617-23
- Fu J, Bian M, Xin G, Deng Z, Luo J, Guo X, Chen H, Wang Y, Jiang Q, Zhang C (2015) TPX2 phosphorylation maintains metaphase spindle length by regulating microtubule flux. *J Cell Biol* 210: 373-83
- Polverino F, Naso FD, Asteriti IA, Palmerini V, Singh D, Valente D, Bird AW, Rosa A, Mapelli M, Guarguaglini G (2021) The Aurora-A/TPX2 Axis Directs Spindle Orientation in Adherent Human Cells by Regulating NuMA and Microtubule Stability. *Current Biology* 31: 658-667.e5
- Zhang D, Tang Z, Huang H, Zhou G, Cui C, Weng Y, Liu W, Kim S, Lee S, Perez-Neut M, Ding J, Cxyz D, Hu R, Ye Z, He M, Zheng YG, Shuman HA, Dai L, Ren B, Roeder RG et al. (2019) Metabolic regulation of gene expression by histone lactylation. *Nature* 574: 575-580
- Chen Y, Wu J, Zhai L, Zhang T, Yin H, Gao H, Zhao F, Wang Z, Yang X, Jin M, Huang B, Ding X, Li R, Yang J, He Y, Wang Q, Wang W, Kloeber JA, Li Y, Hao B et al. (2024b) Metabolic regulation of homologous recombination repair by MRE11 lactylation. *Cell* 187: 294-311.e21

January 27, 2025

Re: Life Science Alliance manuscript #LSA-2024-02978-TR

Xiuying Zhong
University of Science and Technology of China
School of Life Science
Hefei, Anhui 230027
China

Dear Dr. Zhong,

Thank you for submitting your revised manuscript entitled "TPX2 lactylation is required for the cell cycle regulation and hepatocellular carcinoma progression" to Life Science Alliance. The manuscript has been seen by the original reviewers whose comments are appended below. While the reviewers continue to be overall positive about the work in terms of its suitability for Life Science Alliance, some important issues remain.

Our general policy is that papers are considered through only one revision cycle; however, given that the suggested changes are relatively minor, we are open to one additional short round of revision. Please note that I will expect to make a final decision without additional reviewer input upon re-submission.

Please submit the final revision within one month, along with a letter that includes a point by point response to the remaining reviewer comments.

To upload the revised version of your manuscript, please log in to your account: <https://lsa.msubmit.net/cgi-bin/main.plex>
You will be guided to complete the submission of your revised manuscript and to fill in all necessary information.

B. MANUSCRIPT ORGANIZATION AND FORMATTING:

Sincerely,

Reviewer #1 (Comments to the Authors (Required)):

In the revision phase, Liu and colleagues have performed several experiments in response to most of the points I had raised.

Still, only a minor fraction of these new observations has been included in the manuscript, and most of the times the reason is not clearly explained. Additionally, some points that had been raised have been only partially addressed. Therefore, in my opinion some conclusions are not supported by results. Overall, I regard that the manuscript should be further integrated with results shown in the Figures for reviewers and with the deriving discussion, and that some of the conclusions should be mitigated, as detailed below.

- Comments 1-2 and 1-3: authors have verified that TPX2 lactylation increases along with its levels during cell cycle progression. I feel this is an important piece of information that should be included. The same is true for the FACS analyses under the different treatment conditions, shown in R1, which should be included as supplementary material. Indeed, these controls rule out the possibility that lactylation changes are due to cell cycle variations, while at the same time showing that a cell cycle effect is induced.

Here, and elsewhere in Figures, I have an issue with the ratio of "lactylated TPX2/TPX2" in the blots. a. When this is calculated on a TPX2 IP, the ratio should be directly calculated between the TPX2 IPed amount and the lactylation signal. No normalization for actin is required because normalization is made on the specific signal itself. b. The same applies when authors make the ratio between the TPX2 amount in the K1a IP with respect to the total amount of TPX2 in the extract. Dividing for the TPX2 signal in the input already provides the normalization factor, with no need for actin normalization. Introducing the actin normalization step, unless applied to both signals (which would make the correction useless because of the mathematical formula of the ratio), may modify the real ratio. For all ratios, can the standard deviation (deriving from the 3 replicates) be provided? Finally, the performed quantification of TPX2 and its modified form in non transformed and cancer cell lines should be included in the main or supplementary figures.

- Comments 1-6: authors have added the information that TPX2 mutant mitotic localization is not altered and that associated AurkA localization is also comparable to controls; I would include also the nuclear localization shown for Reviewers only. The observation about defective spindle morphology is interesting but is not addressed (by characterizing the defect, or at least providing a quantification); therefore, based on one image in the supplementary material, I do not feel that conclusions on spindle assembly can be drawn. Text should be rephrased to suggest this as an interesting observation for future investigations. I would also avoid "maturation of spindle assembly", since the maturation process rather refers to the centrosome.

- Comments 1-7 and 1-8: as stated above, the analysis of spindle structure cannot be based on the example image provided in the supplementary material. Additionally, no time lapse analysis, mitotic index or mitotic progression have been performed under the conditions shown in Figure 4. Therefore, in my opinion the provided data support the conclusion that a delay in cell cycle progression is induced, during the G2 and M phases (which cannot be distinguished in FACS analyses). There is no evidence of an arrest and there is no evidence of a problem at the "G2/M transition", since G2 and M phases (and hence the transition from one to the other) cannot be distinguished. Conclusions should be modified accordingly throughout the text and the expressions "G2/M arrest" and "G2/M transition" avoided (text and Figures).

- Comments 1-9: data in figure R6 should be included and discussed.

- Comments 1-11: I feel the fact that TPX2 is not detected in the supernatant is an interesting indication that most of it is lactylated. I am not sure why authors regard this observation not to be shown. Concerning the ratio of signals, please see comment 1-2 and 1-3.

- Comments 1-12: controls in R8 should be included.

- Discussion: in the sentence "further work will be required to generate inhibitors that could specifically TPX2 K249 lactylation" one word must be missing.

- Finally, I find the final scheme slightly confusing: authors show in the manuscript that AurkA/TPX2 binding is not affected by lactylation (or lack of lactylation). Why is the de-lactylated TPX2 shown as unbound from AurkA in the right part of the scheme? Also, since the left and right part of the scheme refer to a physiological situation of a reversible modification, I find misleading to assimilate physiologically unmodified TPX2 to the described mutant, particularly in terms of downstream events (G2/M arrest, tumor suppression).

Reviewer #2 (Comments to the Authors (Required)):

In general, these authors have addressed my comments or concerns appropriately, and the revised manuscript may be considered for acceptance and publication

Point-by-point response to the comments

Editor's Decision:

Thank you for submitting your revised manuscript entitled "TPX2 lactylation is required for the cell cycle regulation and hepatocellular carcinoma progression" to Life Science Alliance. The manuscript has been seen by the original reviewers whose comments are appended below. While the reviewers continue to be overall positive about the work in term of its suitability for Life Science Alliance, some important issues remain.

Our general policy is that papers are considered through only one revision cycle; however, given that suggested changes are relatively minor, we are open to one additional short round of revision. Please note that I will expect to make a final decision without additional reviewer input upon re-submission.

Please submit the final revision within one month, along with a letter that includes a point-by-point response to the remaining reviewer comments.

Response: We are grateful for your kind decision allowing us to revise the manuscript. We also thank the reviewers for their insightful comments and suggestions. We believe that all the comments are critical and relevant and helped us substantially improve the study.

Over the past one month, we have made improvements to address all the concerns and comments raised by our reviewers. We are now submitting a significantly improved manuscript along with our point-by-point response.

Point-by-point response to the comments of the Reviewers

Reviewer #1:

Comments 1-1:

In the revision phase, Liu and colleagues have performed several experiments in response to most of the points I had raised. Still, only a minor fraction of these new observations has been included in the manuscript, and most of the times the reason is not clearly explained. Additionally, some points that had been raised have been only partially addressed. Therefore, in my opinion some conclusions are not supported by results. Overall, I regard that the manuscript should be further integrated with results shown in the Figures for reviewers and with the deriving discussion, and that some of the conclusions should be mitigated, as detailed below.

Response: We thank the reviewer for the specific and insightful concerns, which we will address one by one below.

Comments 1-2:

Comments 1-2 and 1-3: authors have verified that TPX2 lactylation increases along with its levels during cell cycle progression. I feel this is an important piece of information that should be included. The same is true for the FACS analyses under the different treatment conditions, shown in R1, which should be included as supplementary material. Indeed, these controls rule out the possibility that lactylation changes are due to cell cycle variations, while at the same time showing that a cell cycle effect is induced.

Here, and elsewhere in Figures, I have an issue with the ratio of "lactylated TPX2/TPX2" in the blots. a. When this is calculated on a TPX2 IP, the ratio should be directly calculated between the TPX2 IPed amount and the lactylation signal. No normalization for actin is required because normalization is made on the specific signal itself. b. The same applies when authors make the ratio between the TPX2 amount in the K1a IP with respect to the total amount of TPX2 in the extract. Dividing for the TPX2 signal in the input already provides the normalization factor, with no need for actin normalization. Introducing the actin normalization step, unless applied to both signals (which would make the correction useless because of the mathematical formula of the ratio), may modify the real ratio. For all ratios, can the standard deviation (deriving from the 3 replicates) be provided? Finally, the performed quantification of TPX2 and its modified form in non-transformed and cancer cell lines should be included in the main or supplementary figures.

Response: We appreciate the reviewer’s insightful comments. Following the reviewer’s suggestions, we have included the data showing the changes of TPX2 lactylation during cell cycle as **Figure S4A** in the revised manuscript. Our FACS data that the reviewer mentioned showed that decreased lactate level led to delayed cell cycle, which is consistent with the previous study (Liu *et al.*, 2023). Since the role of lactate in regulating the cell cycle has been addressed in the study, we have regretfully decided not to include these results in the revised manuscript.

We also greatly appreciate the reviewer’s valuable comments regarding the method used to calculate the ratio of TPX2 lactylation. Following reviewer’s suggestions, we have recalculated the ratio of TPX2 lactylation without Actin normalization in **Figure 1I, 1J, 2A** and **2E** in the revised manuscript. Additionally, we also have presented the standard deviation of TPX2 lactylation based on three independent experiments. For your convenience, we have appended these results here as **Figure R1-R4**. Following the reviewer’s suggestions, we have included the data showing the levels of TPX2 lactylation in HCC cell lines and non-transformed cell line THLE3 as **Figure 1I** in the revised manuscript.

Figure R1. TPX2 lactylation was induced in HCC cells.

(A) Western blot analysis of the lactylation of TPX2 in the indicated cell lines using IP samples as indicated. Blot bands were quantified by ImageJ and the fraction of lactylated TPX2 were represented by the ratio of the Kla band to the immunoprecipitated TPX2 band. The data were presented as the mean ± SD of three independent experiments, n=3

Figure R2. TPX2 is lactylated by CBP.

(A) Western blot analysis of the lactylation of TPX2 in HEK293T cells with Flag-TPX2 and lactylase overexpression using IP samples as indicated. Blot bands were quantified by ImageJ and the fraction of lactylated TPX2 were represented by the ratio of the Klalactylated TPX2 to the immunoprecipitated Flag-TPX2 band. The data were presented as the mean \pm SD of three independent experiments, n=3.

Figure R3. TPX2 was delactylated by HDAC1.

(A) Western blot analysis of the lactylation of TPX2 in HEK293T cells with Flag-TPX2 and delactylase overexpression using IP samples as indicated. Blot bands were quantified by ImageJ and the fraction of lactylated TPX2 were represented by the ratio of the K1a band to the immunoprecipitated Flag-TPX2 band. The data were presented as the mean \pm SD of three independent experiments, n=3.

Figure R4. TPX2 lactylation gradually increased from G1 to G2/M phase.

(A) Western blot analysis of the lactylation of TPX2 in Hep3B cells using IP samples as indicated. Hep3B cells were synchronized using a double-thymidine block and then cells were collected immediately (G1 phase) or released to complete DMEM containing 10% FBS for 6 h (S phase) or 10 h (G2/M phase). Blot bands were quantified by ImageJ and the fraction of lactylated TPX2 were represented by the ratio of the K1a band to the immunoprecipitated TPX2 band. The data were presented as the mean \pm SD of three independent experiments, n=3.

Comments 1-3:

Comments 1-6: authors have added the information that TPX2 mutant mitotic localization is not altered and that associated AurKA localization is also comparable to controls; I would include also the nuclear localization shown for Reviewers only. The observation about defective spindle morphology is interesting but is not addressed (by characterizing the defect, or at least providing a quantification); therefore, based on one image in the supplementary material, I do not feel that conclusions on spindle assembly can be drawn. Text should be rephrased to suggest this as an interesting observation for future investigations. I would also avoid "maturation of spindle assembly", since the maturation process rather refers to the centrosome.

Response: We thank the reviewer for the insightful comments. First, following the reviewer's suggestion, we have appended the interphase nuclear localization of TPX2 and AURKA under different ectopic transgenic TPX2 expressing conditions in **Figure S4F** in the revised manuscript. We also appreciate the reviewer for interest in our finding that TPX2 lactylation may affect the mitotic spindle morphology and pointing out that more sufficient evidences would be needed to address this observation. We totally agree with the reviewer. Therefore, we take the advice of the reviewer to rephrase to suggest this interesting observation warrants further independent study in the revised manuscript.

Comments 1-4:

Comments 1-7 and 1-8: as stated above, the analysis of spindle structure cannot be based on the example image provided in the supplementary material. Additionally, no time lapse analysis, mitotic index or mitotic progression have been performed under the conditions shown in Figure 4. Therefore, in my opinion the provided data support the conclusion that a delay in cell cycle progression is induced, during the G2 and M phases (which cannot be distinguished in FACS analyses). There is no evidence of an arrest and there is no evidence of a problem at the "G2/M transition", since G2 and M phases (and hence the transition from one to the other) cannot be distinguished. Conclusions should be modified accordingly throughout the text and the expressions "G2/M arrest" and "G2/M transition" avoided (text and Figures).

Response: We thank the reviewer for the critical comments. We totally agree the reviewer that the defective spindle morphology in TPX2 lactylation-deficient cells is an interesting observation and additional evidence should be provided. We have tried in the past one month, but it's impossible for us to accomplish time lapse analysis, and the analysis of mitotic index or mitotic progression during the time frame allowed for this

revision. In addition, considering the potential roles of TPX2 lactylation in regulating spindle morphology might be important not only for tumour cells but also for other fast proliferating cells, we think further independent study is warranted. Therefore, following the reviewer's suggestion, we rephrase to state that TPX2 lactylation is necessary for the cell cycle regulation in HCC cells, avoiding the terms "G2/M arrest" and "G2/M transition" in the revised manuscript.

Comments 1-5:

Comments 1-9: data in figure R6 should be included and discussed.

Response: We truly appreciate the reviewer for the comments. In the first revision, following reviewer's suggestions, we added control groups using parental cells with no manipulation of TPX2 levels in the WB assay and FACS analysis in **Figure R6**, which undoubtedly further validated the effects of the inhibitors used in the experiments. Actually, as the reviewer mentioned in the first revision, the effects of Alisertib on cell viability and tumor progression has been well studied (Zhu *et al.*, 2017, Lin *et al.*, 2024). In addition, according to the study of Chen *et al.*, inhibition of lactate production might be a promising therapeutic cancer strategy (Chen *et al.*, 2024). In the present study, we focused on revealing the critical roles of lactate-induced TPX2 lactylation in regulating cell cycle and promoting cancer progression and the underlining mechanisms. Considering that the inhibitors had similar effects on parental cells and NTC+EV (non-targeting control and empty vector) control cells, we did not include these results using parental cells as control in the revised manuscript. We agree with the reviewer's opinion that lactylation is relatively newly identified posttranslational modification, and have discussed the potential roles of TPX2 lactylation and LDH inhibitors for new cancer interventions in the manuscript.

Comments 1-6:

Comments 1-11: I feel the fact that TPX2 is not detected in the supernatant is an interesting indication that most of it is lactylated. I am not sure why authors regard this observation not to be shown. Concerning the ratio of signals, please see comment 1-2 and 1-3.

Response: We truly appreciate the reviewer for the advice. Following reviewer's suggestion, we have recalculated the ratio of TPX2 lactylation. We also appended the results of three independent experiments here as **Figure R5**. The quantification of TPX2 lactylation in IP samples validated that large fraction of TPX2 was lactylated in HCC cells. Due to the low level of TPX2 in the supernatant, the signals in the

supernatant were too faint to accurately calculate the corresponding ratio. Therefore, we did not include the results of TPX2 detection in supernatant in the manuscript.

A

Figure R5. TPX2 was lactylated in HCC cells.

(A) Western blot analysis of the lactylation of TPX2 in HepG2 cells using IP samples as indicated. Blot bands were quantified by ImageJ and the fraction of lactylated TPX2 were represented by the ratio of the immunoprecipitated TPX2 band to the corresponding Input TPX2 band. The data were presented as the mean \pm SD of three independent experiments, $n=3$.

Comments 1-7:

Comments 1-12: controls in R8 should be included.

Response: We are very grateful to the reviewer for the valuable advice. Following reviewer's suggestion, we have performed a control experiment to clarify that the treatments in our study effectively regulate the pan-lactylation levels in cells. However, this experimental treatment is a well-established method of modulating lactylation in the cells (Zhang *et al.*, 2019, Yu *et al.*, 2021, Zhu *et al.*, 2025). Therefore, we did not include this experiment control in the revised manuscript.

Comments 1-8:

Discussion: in the sentence "further work will be required to generate inhibitors that could specifically TPX2 K249 lactylation" one word must be missing.

Response: We are grateful for the reviewer's insightful comments. The mistake sentence is corrected as "further work will be required to generate inhibitors that could specifically suppress TPX2 K249 lactylation".

Comments 1-9:

Finally, I find the final scheme slightly confusing: authors show in the manuscript that AurkA/TPX2 binding is not affected by lactylation (or lack of lactylation). Why is the de-lactylated TPX2 shown as unbound from AurkA in the right part of the scheme? Also, since the left and right part of the scheme refer to a physiological situation of a reversible modification, I find misleading to assimilate physiologically unmodified TPX2 to the described mutant, particularly in terms of downstream events (G2/M arrest, tumor suppression).

Response: We thank the reviewer for the critical comments. Following reviewer's suggestions, we have revised the schematic in the manuscript to more clearly illustrate the underlying mechanism observed in our study. TPX2 lactylation is induced in tumor cells dependent on lactate production and lactylase CBP, which is essential for its role in protecting AURKA T288 phosphorylation from phosphatase PP1 and promoting tumour growth. When TPX2 is delactylated by the inhibition of lactate production or delactylase overexpression, the delactylated TPX2 cannot prevent the interaction between PP1 and AURKA, leading to AURKA T288 dephosphorylation and delayed cell cycle. In conclusion, our results reveal that TPX2 lactylation promotes HCC progression by disrupting the interaction between AURKA and PP1 to facilitate AURKA activation and cell cycle progression.

Reviewer #2:

Comments 2-1:

In general, these authors have addressed my comments or concerns appropriately, and the revised manuscript may be considered for acceptance and publication.

Response: We appreciate the reviewer for the positive opinion of our work.

References

- Liu W, Wang Y, Bozi LHM, Fischer PD, Jedrychowski MP, Xiao H, Wu T, Darabedian N, He X, Mills EL, Burger N, Shin S, Reddy A, Sprenger HG, Tran N, Winther S, Hinshaw SM, Shen J, Seo HS, Song K et al. (2023) Lactate regulates cell cycle by remodelling the anaphase promoting complex. *Nature* 616: 790-797
- Zhu Q, Yu X, Zhou Z-W, Zhou C, Chen X-W, Zhou S-F (2017) Inhibition of Aurora A Kinase by Alisertib Induces Autophagy and Cell Cycle Arrest and Increases Chemosensitivity in Human Hepatocellular Carcinoma HepG2 Cells. *Current Cancer Drug Targets* 17: 386-401
- Lin X, Pan F, Abudoureyimu M, Wang T, Hao L, Wang R (2024) Aurora-A inhibitor synergistically enhances the inhibitory effect of anlotinib on hepatocellular carcinoma. *Biochemical and Biophysical Research Communications* 690
- Chen H, Li Y, Li H, Chen X, Fu H, Mao D, Chen W, Lan L, Wang C, Hu K, Li J, Zhu C, Evans I, Cheung E, Lu D, He Y, Behrens A, Yin D, Zhang C (2024) NBS1 lactylation is required for efficient DNA repair and chemotherapy resistance. *Nature* 631: 663-669
- Zhang D, Tang Z, Huang H, Zhou G, Cui C, Weng Y, Liu W, Kim S, Lee S, Perez-Neut M, Ding J, Czyn D, Hu R, Ye Z, He M, Zheng YG, Shuman HA, Dai L, Ren B, Roeder RG et al. (2019) Metabolic regulation of gene expression by histone lactylation. *Nature* 574: 575-580
- Yu J, Chai P, Xie M, Ge S, Ruan J, Fan X, Jia R (2021) Histone lactylation drives oncogenesis by facilitating m(6)A reader protein YTHDF2 expression in ocular melanoma. *Genome Biol* 22: 85
- Zhu R, Ye X, Lu X, Xiao L, Yuan M, Zhao H, Guo D, Meng Y, Han H, Luo S, Wu Q, Jiang X, Xu J, Tang Z, Tao YJ, Lu Z (2025) ACSS2 acts as a lactyl-CoA synthetase and couples KAT2A to function as a lactyltransferase for histone lactylation and tumor immune evasion. *Cell Metab* 37: 361-376 e7

February 27, 2025

RE: Life Science Alliance Manuscript #LSA-2024-02978-TRR

Xiuying Zhong
University of Science and Technology of China
School of Life Science
Hefei, Anhui 230027
China

Dear Dr. Zhong,

Thank you for submitting your revised manuscript entitled "TPX2 lactylation is required for the cell cycle regulation and hepatocellular carcinoma progression". We would be happy to publish your paper in Life Science Alliance pending final revisions necessary to meet our formatting guidelines.

- please be sure that the authorship listing and order is correct
- please add the Twitter handle of your host institute/organization as well as your own or/and one of the authors in our system
- please use the [10 author names et al.] format in your references (i.e., limit the author names to the first 10)
- please rename the list of the supplementary figure legends in the manuscript text from Expanded View Figures legends to Supplementary Figure legends, not Expanded View Figures legends
- Please remove panel A from the legend, the figure itself, and the callout in the text for Figure S5. There is no need to label the panel since the figure has only one panel
- please remove the Highlights section from page 2

LSA now encourages authors to provide a 30-60 second video where the study is briefly explained. We will use these videos on social media to promote the published paper and the presenting author (for examples, see <https://docs.google.com/document/d/1-UWCfbE4pGcDdcgzcmiuJI2XMBJnxKYeqRvLLrLS08s/edit?usp=sharing>). Corresponding or first-authors are welcome to submit the video. Please submit only one video per manuscript. The video can be emailed to contact@life-science-alliance.org

A. FINAL FILES:

B. MANUSCRIPT ORGANIZATION AND FORMATTING:

Thank you for your attention to these final processing requirements. Please revise and format the manuscript and upload materials within 5 days.

Sincerely,

March 3, 2025

RE: Life Science Alliance Manuscript #LSA-2024-02978-TRRR

Xiuying Zhong
South China University of Technology
School of Life Science
Guangzhou
Guangzhou, GuangDong 510006
China

Dear Dr. Zhong,

Thank you for submitting your Research Article entitled "TPX2 lactylation is required for the cell cycle regulation and hepatocellular carcinoma progression". It is a pleasure to let you know that your manuscript is now accepted for publication in Life Science Alliance. Congratulations on this interesting work.

DISTRIBUTION OF MATERIALS:

Again, congratulations on a very nice paper. I hope you found the review process to be constructive and are pleased with how the manuscript was handled editorially. We look forward to future exciting submissions from your lab.

Sincerely,
